# An improved neuroanatomical model of the default-mode network reconciles previous neuroimaging and neuropathological findings

Pedro Nascimento Alves[1,2,3,4]*, Chris Foulon [1,2,5], Vyacheslav Karolis [1,2,6], Danilo Bzdok [7,8,9,10], Daniel S. Margulies[1,2], Emmanuelle Volle[1,2] & Michel Thiebaut de Schotten [1,2,11,12]*

The brain is constituted of multiple networks of functionally correlated brain areas, out of which the default-mode network (DMN) is the largest. Most existing research into the DMN has taken a corticocentric approach. Despite its resemblance with the unitary model of the limbic system, the contribution of subcortical structures to the DMN may be under-appreciated. Here, we propose a more comprehensive neuroanatomical model of the DMN including subcortical structures such as the basal forebrain, cholinergic nuclei, anterior and mediodorsal thalamic nuclei. Additionally, tractography of diffusion-weighted imaging was employed to explore the structural connectivity, which revealed that the thalamus and basal forebrain are of central importance for the functioning of the DMN. The contribution of these neurochemically diverse brain nuclei reconciles previous neuroimaging with neuropathological findings in diseased brains and offers the potential for identifying a conserved homologue of the DMN in other mammalian species.

[1] Brain Connectivity and Behaviour Laboratory, BCBlab, Sorbonne Universities, Paris, France. [2] Frontlab, Institut du Cerveau et de la Moelle épinière (ICM), UPMC UMRS 1127, Inserm U 1127, CNRS UMR 7225 Paris, France. [3] Department of Neurosciences and Mental Health, Neurology, Hospital de Santa Maria, CHULN, Lisbon, Portugal. [4] Language Research Laboratory, Faculty of Medicine, Universidade de Lisboa, Lisbon, Portugal. [5] Computational Neuroimaging Laboratory, Department of Diagnostic Medicine, The University of Texas at Austin Dell Medical School, Austin, TX, USA. [6] FMRIB centre, John Radcliffe Hospital, University of Oxford, Oxford, UK. [7] INRIA, Parietal Team, Saclay, France. [8] Neurospin, CEA, Gif-sur-Yvette, France. [9] Department of Psychiatry, Psychotherapy and Psychosomatics, RWTH Aachen University, Aachen, Germany. [10] JARA-BRAIN, Jülich-Aachen Research Alliance, Jülich, Germany. [11] Centre de Neuroimagerie de Recherche CENIR, Groupe Hospitalier Pitié-Salpêtrière, Paris, France. [12] Groupe d'Imagerie Neurofonctionnelle, Institut des Maladies Neurodégénératives-UMR 5293, CNRS, CEA University of Bordeaux, Bordeaux, France. *email: pedronascimentoalves@gmail.com; michel.thiebaut@gmail.com

In 1979, for the first time to our knowledge, David Ingvar used Xenon clearance to investigate resting wakefulness[1]. When aligned by scalp and skull markers, the 11 brains examined indicated an evident increase of the blood flow levels in the frontal lobe interpreted as a surrogate for undirected, spontaneous, conscious mental activity. Later, positron emission tomography (PET) was used to map more systematically task-related activation in the brain, often with resting wakefulness as a control task. The contrast between task-related and resting wakefulness led to the description of deactivation (i.e., active at rest more than during the task) in a set of regions, including retrosplenial cortex, inferior parietal cortex, dorsolateral frontal cortex inferior frontal cortex, left inferior temporal gyrus, medial frontal regions and amygdala[2,3] that quickly bore the name of default-mode network (DMN)[4]. In these studies, skull landmarks or structural magnetic resonance imaging (MRI) were used to align PET images in Talairach stereotaxic or in Montreal Neurological Institute (MNI) templates[2,3]. The advent of functional magnetic resonance imaging (fMRI), particularly of methods for analysing functional connectivity, led to the allocation of new structures to this network, such as the hippocampal formation[5–7].

Today, the DMN has largely been a cortically defined set of network nodes. Consisting of distinct regions/nodes distributed across the ventromedial and lateral prefrontal, posteromedial and inferior parietal, as well as the lateral and medial temporal cortex, the DMN is considered a backbone of cortical integration[8–11]. Its subcortical components are, however, less well characterised. Studies of whole-brain network organisation reveal subregions of the cerebellum[12,13] and striatum[14] that are functionally connected with the cortical regions of the DMN. Seed-based functional connectivity studies further demonstrate additional DMN-specific connectivity to several subcortical structures, including the amygdala[15,16] and striatum[17]. The thalamus has also been shown to be structurally and functionally connected to DMN regions[18,19]. These studies are important, as a cleaner characterisation of the anatomy of the DMN is an essential step towards understanding its functional role and its involvement in brain diseases. Particularly, an increased activity characterises the regions that compose DMN during tasks involving autobiographical, episodic and semantic memory, mind wandering, perspective-taking or future thinking[20,21]; Bendetowicz et al.[22]. Conversely, DMN regions show a decreased neural activity during attention-demanding and externally oriented tasks[21,23]. Finally, altered connectivity in the DMN has been observed in a large variety of brain diseases, including Alzheimer's disease, Parkinson's disease, schizophrenia, depression, temporal lobe epilepsy, attention deficit and hyperactivity disorder, drug addiction, among others[24–29]. Hence, while prior research provides first hints towards a broader definition of the DMN system, further research is necessary to articulate the anatomical extent of specific subcortical contributions, and to understand the independent contribution of these structures in DMN function and pathologies.

Yet, since the DMN has repeatedly been characterised as a cohesive functional network[5], an average of brain images relying exclusively on anatomical references and landmarks may be suboptimal[30,31] whether the method employed is a surface-based or volume-based registration[30,32]. Small structures of the brain may be particularly susceptible to this misalignment, especially when MRI lacks contrast. Besides morphology, cytoarchitecture and function are poorly overlapping, especially in the DMN[33,34]. Consequently, functional areas present in every subject may not overlap after averaging all structurally aligned brain images in a group analysis[30,35]. This biological misalignment can be particularly problematic for revealing significant small regions of the DMN (Fig. 1). A better alignment is also essential for the subcortical structures of the brain, as their variability is still considerable[36–39]. Specifically, cytoarchitectonic studies have shown that only one-quarter of the volume of cholinergic nuclei overlaps in at least half of the individuals studied[39]. Similarly, structures such as mammillary bodies, nucleus basalis of Meynert, or anterior thalamic nuclei can vary in size, morphology and locations, and are particularly prone to misalignment with the current methods of structural registration[32,40–42]. Functional alignment methods have already been used to overcome the high interindividual variability of the morphology of some areas of the heteromodal association cortex and led to a more accurate mapping of resting-state functional connectivity[43–45]. They also have led to better predictions of task activation patterns in group analysis when compared with morphological alignment methods and have recently enabled much improved models of cortical parcellation[43,46].

To address several of these challenges that may impede proper group analysis if relying exclusively on MRI structural landmarks, we propose to revisit the anatomical scaffold of the DMN using a coregistration based on functional alignment. We hypothesised that using a functional alignment will reveal structures of basal forebrain and the Papez's circuits, namely anterior and mediodorsal thalamic nuclei and mammillary bodies, as constituent nodes of the DMN for several reasons. First, all these regions are highly interconnected which suggest they belong to the same functional system[47,48]. Second, the current conceptualisation of DMN anatomy resembles the unitary model of the limbic system which, through the coordination of its subregions, subserves the elaboration of emotion, memories and behaviour[49–52]. Third, the basal forebrain comprises a group of neurochemically diverse nuclei, involved in dopaminergic, cholinergic and serotoninergic pathways, that are crucial in the pathophysiology of the aforementioned diseases that affect the DMN connectivity. Finally, recent electrophysiological evidence has shown that in rats the basal forebrain exhibits the same pattern of gamma oscillations than DMN and that it influences the activity of the anterior cingulate cortex[53].

Therefore, in this study, we used a functional alignment of resting-state functional MRI (rs-fMRI)-based individual DMN maps to build a more comprehensive DMN model that includes the contribution of subcortical structures. To provide a complete window into the anatomy of the DMN, we explored the structural connectivity of our new model of the DMN using tractography imaging techniques and revealed that the thalamus and basal forebrain had high importance in term of values of node degree and centrality for the DMN.

## Results

**Comparison between structural and functional alignment.** DMN connectivity maps obtained from structural and functional alignmentsare displayed in Fig. 2a–c. In both maps, classical areas of the DMN were observed, namely: posterior cingulate cortex and retrosplenial cortex; ventromedial, anteromedial and dorsal prefrontal cortex; temporal pole; middle temporal gyrus; hippocampus and parahippocampal cortex; amygdala and the posterior parietal cortex.

Figure 2d illustrates the simple voxel-based subtraction between the DMN connectivity maps obtained from structural and functional alignments. Higher average connectivity was achieved in the functionally aligned DMN map in large areas, such as the medial prefrontal cortex and posterior cingulate cortex, mostly in the border zones. In fact, the highest differences in connectivity between structural and functional aligned DMN were at the level of the basal forebrain and thalamus. These areas were poorly or even not represented with alignment in the

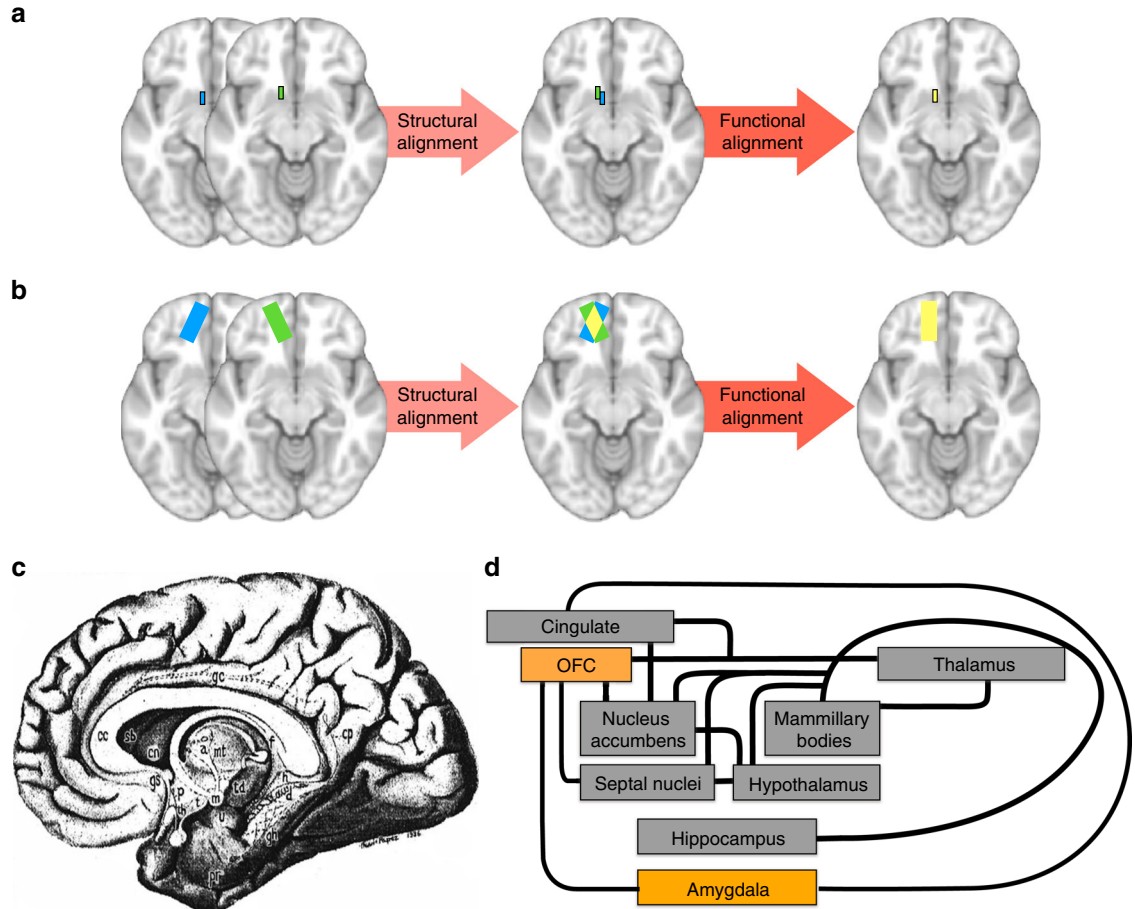

**Fig. 1** Illustration of intersubject alignment of brain images and unitary model of the limbic system. Blue and green rectangles represent the same functional area in two subjects, while yellow rectangles illustrate the overlap of the two individual areas after alignment. With structural alignment, there can be a complete misalignment of small functional areas (**a**) or partial misalignment of large functional areas (**b**) due to functional–anatomical variability or poor anatomical contrasts in MRI imaging. If an additional step of functional alignment is performed, an optimised overlap of functional areas is obtained. **c** The limbic system as originally depicted by Papez[52] and (**d**) diagram of the unitary model of the limbic system[49–51,165–168]

MNI152 space, but were visible after the functional alignment. DMN connectivity was also visible in the medial mesencephalic region, as well in inferior regions in the caudate nuclei, ventrolateral prefrontal cortex, cerebellar tonsils and cerebellar hemispheres. As expected, a difference was found between the two maps bilaterally in the thalamus and in the basal forebrain, and in a peripheral zone of the left mammillary body (Fig. 2e, Table 1). Unthresholded statistical maps of the comparison between the two methods of alignment—structurally and functionally aligned DMN—are available at neurovault.org (https://neurovault.org/collections/OCAMCQFK/). Two representative cases of individual maps in the MNI152 space and in the functional space are presented in the Supplementary Fig. 1.

**Functional connectivity of the DMN in the functional space.** Twenty-four regions of interest were defined based on the DMN we obtained in the functional space and concordant with the previous anatomical models of the DMN[5,8]. Nine additional regions were defined as being solely revealed after the functional alignment. These regions included the left and right thalamus, the left and right basal forebrain, the midbrain, the left and right ventral lateral prefrontal cortex (VLPFC) and the left and right caudate nucleus (inferior regions of the nuclei), resulting in a total of 33 regions of interest (Supplementary Fig. 2). The association strength determined by Pearson's correlation between the rs-

*f*MRI time series of the regions of interest (i.e., functional connectivity) were higher with alignment in the functional space, compared with structural space, in all pairs of regions (Fig. 3a). The difference was statistically significant in 18% of pairs after Bonferroni correction for multiple comparisons (*p*-value < 0.0001). Table 2 represents the MNI coordinates of the centres of gravity of all the regions of interest in the DMN map. The alternative correlation matrices using cortical DMN areas according to Gordon et al.[54] are presented in the Supplementary Fig. 3.

Regarding the hypothesised areas, the left basal forebrain demonstrated higher correlations with the right antero-median prefrontal cortex, the right posterior parietal cortex and the midbrain area, while the right basal forebrain had higher correlations with the right temporal pole and with the left cerebellar hemisphere (Supplementary Data 1). The left thalamus had higher correlations with the left parahippocampal region, the left temporal pole, the right and the left antero-median prefrontal cortex, the left and the right ventrolateral prefrontal cortex and the right posterior parietal cortex, while the right thalamus had higher correlations with the right and the left antero-median prefrontal cortex, with the left temporal pole and with the left cerebellar tonsil (Supplementary Data 1). No significant difference was found for partial correlations. Tables presenting the median, range and interquartile range of partial correlations between all regions of interests, the statistical comparison of

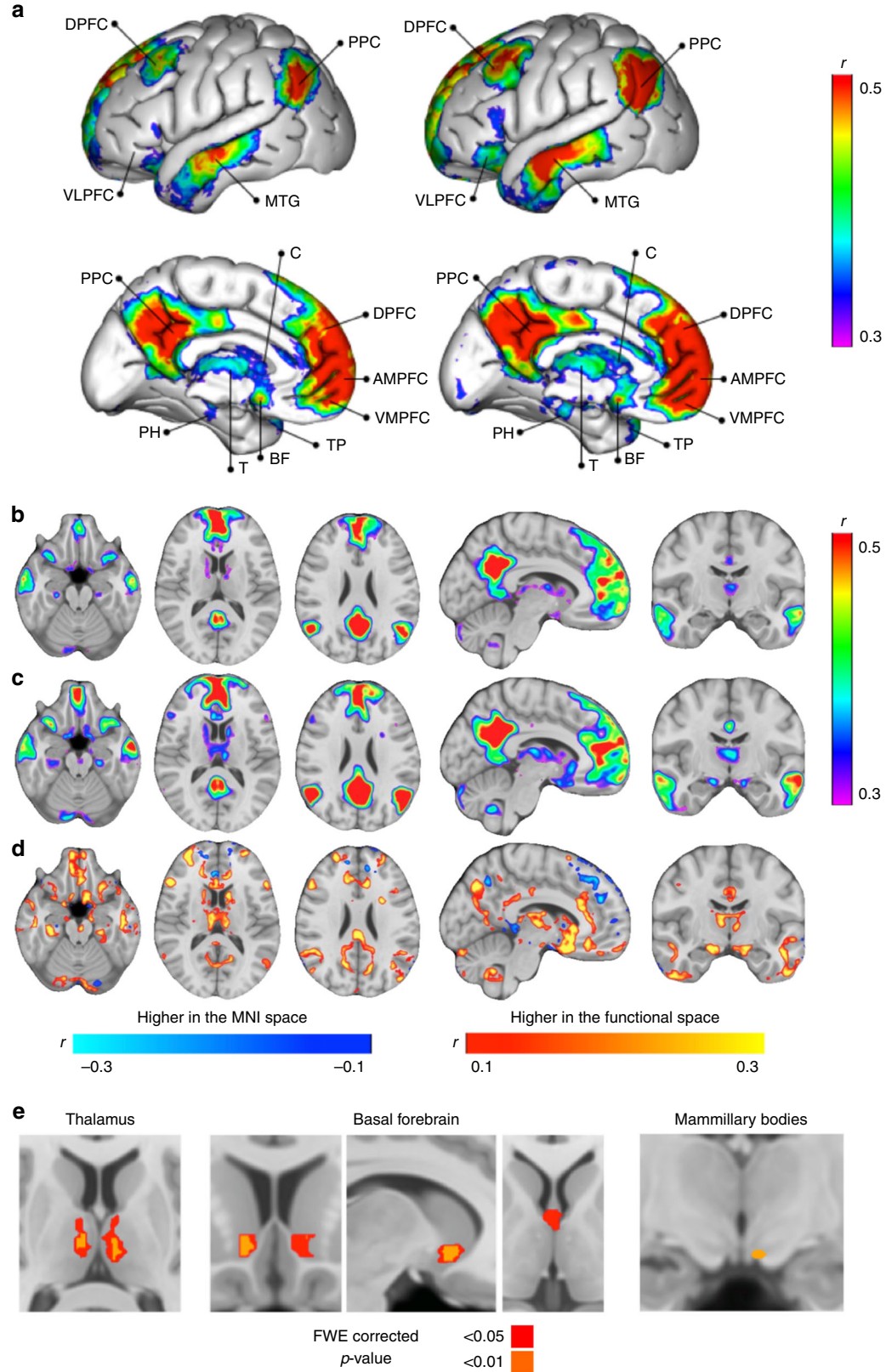

partial correlations between the two methods of alignment and the partial correlations of two representative cases are also shown in the Supplementary Data 2, 3, 4 and 5, respectively.

**Anatomical validation in thalamic, basal forebrain and mesencephalic areas**. Figure 4 illustrates the intersection of the

new DMN map after its translation to the MNI group space using individual inverse transformation matrices specific for each individual. All subjects' DMN spatially overlapped with the templates of the left anterior thalamic nucleus, mediodorsal thalamic nuclei, medial septal nuclei and left nucleus accumbens (Table 3)[39,55–58]. The number of subjects with an intersection

**Fig. 2** Maps of the DMN structurally or functionally aligned. **a** 3D view of the two DMN left panel corresponds to the structural space alignment, right panel to the functional space alignment, **b** brain sections of the structurally aligned DMN, **c** brain sections of the functionally aligned DMN, **d** subtraction of the structurally and the functionally aligned DMN maps, **e** statistical comparison (paired $t$ test) between the two methods of alignment —structurally and functionally aligned DMN—in the three hypothesised regions, with colours indicating statistically significant differences at two levels of significance: <0.05 and <0.01, family-wise error (FWE) corrected $p$-values (higher in the functional space). DPFC dorsal prefrontal cortex, PPC posterior parietal cortex, VLPFC ventrolateral prefrontal cortex, MTG middle temporal gyrus, PCC posterior cingulate cortex, C caudate, DPFC dorsal prefrontal cortex, AMPFC antero-median prefrontal cortex, VMPFC ventro-median prefrontal cortex, TP temporal pole, BF basal forebrain, T thalamus, PH parahippocampal. $n = 20$ participants

---

**Table 1 Clusters of the statistical maps obtained when comparing the two methods of alignment ($t$ test)**

| Cluster index | Voxels | $p$-value | Effect size | MNI ($X$) | MNI ($Y$) | MNI ($Z$) |
|---|---|---|---|---|---|---|
| Right nucleus accumbens | 43 | 0.001 | 0.839 | 9 | 13 | −7 |
| Left nucleus accumbens | 21 | 0.007 | 0.721 | −9 | 16 | −6 |
| Medial septal nuclei | 38 | 0.007 | 0.721 | 0 | −1 | 1 |
| Right limbic thalamus | 72 | 0.001 | 0.839 | 8 | −12 | 9 |
| Left limbic thalamus | 146 | <0.001 | 0.879 | −7 | −14 | 8 |
| Left mammillary bodies | 3 | 0.001 | 0.839 | −5 | −11 | −14 |

Coordinates represent the centre of gravity and '$p$-value' of the lowest value found in the cluster

---

with the right anterior thalamic nucleus, right nucleus accumbens and ventral tegmental area was also very high (95, 95 and 90%, respectively), while the intersection with the other basal forebrain nuclei occurred in approximately half of the subjects, possibly due to their very small size. Maps of the mean and standard deviation Pearson's correlation of the new DMN map after registration to the MNI space are available in the Supplementary Figs. 4 and 5, respectively.

**Tractography**. We explored the structural connectivity of our new model of the DMN using tractography imaging techniques. The regions of interest were the same used for functional connectivity analysis (Supplementary Fig. 2). Figure 5 represents the structural connectivity of the network.

The results indicated that both anterior and posterior portions of the cingulum, as well as inferior longitudinal fasciculus, the second branch of the superior longitudinal fasciculus, the posterior segment of arcuate fasciculus, the uncinate fasciculus and some fibres of the frontal orbito-polar tract (Fig. 5, upper panel) connected the different nodes of the DMN. In addition, the anatomical connectivity of the basal forebrain and the thalamus with other regions of interest included: the anterior thalamic projections, connecting thalamus with medial prefrontal cortex; the cingulum, connecting basal forebrain with medial prefrontal cortex and posterior cingulate cortex; the fornix, connecting basal forebrain (specifically the region correspondent to the medial septal nuclei) to the hippocampus and fibres connecting basal forebrain and thalamus, some of the most medial possibly corresponding to the bundle of Vicq D'Azyr (Fig. 5, lower panel). Statistical maps of the tractography analysis are presented in the Supplementary Fig. 6.

**Graph theory analysis**. Figure 6 represents the analysis of the DMN structural network with a graph theory approach using the 33 regions of interest defined from the DMN in the functional space.

The results indicate that high degrees and high betweenness centrality in the network were obtained for the basal forebrain and thalamic regions, alongside the medial prefrontal cortex and the posterior cingulate–retrosplenial cortex, as well as in regions which were previously considered as hubs in the DMN.

More precisely, the [maximum–minimum] range of distribution of node degrees was [12–1] and the median [interquartile range] was 5 [8–2]. The node degrees of the left and right thalamus were 9 and 7, and of the left and right basal forebrain were 8 and 7, respectively (Supplementary Table 1). Therefore, thalamus and basal forebrain are among the structures in the network that have connections with a high number of nodes. For betweenness centrality, the [maximum–minimum] range of distribution was [0.104–0] and the median [interquartile range] was 0.004 [0.03–0.001]. The betweenness centrality of the left and right thalamus was 0.03, and of the left and right basal forebrain were 0.03 and 0.02, respectively (Supplementary Table 1). Hence, thalamus and basal forebrain make part of a high fraction of shortest paths in the network, that is, the shortest connections between two nodes.

A table showing the values for node degree and betweenness centrality at the individual level in two representative cases is shown in the Supplementary Table 2.

## Discussion

In this study, we revisited the constituent elements of the DMN using an optimised method of coregistration in a functional space, besides the conventional structural alignment. Three main findings emerge from this research in healthy humans. First, higher functional connectivity correlation and sharper anatomical details were achieved when registering the DMN maps in a functional space. Second, we confirmed the hypothesis that structures of basal forebrain and anterior and mediodorsal thalamic nuclei belong to the DMN. Lastly, we characterised in detail the structural connectivity underlying functional connectivity. Based on these findings, we provided a more comprehensive neurobiological model of the DMN that bridges the gap between local differences in subcortical structures and global differences in the DMN reported in clinical studies.

The difference between alignment in the functional space and the structural space was characterised by an increase in the connectivity strength across the brain, as well as in many subcortical areas, classically not considered to be constituent nodes of the DMN. As previously reported, this confirmed that registration in the functional space provides a more accurate interindividual anatomical description and is recommended when doing functional connectivity analyses[43,44].

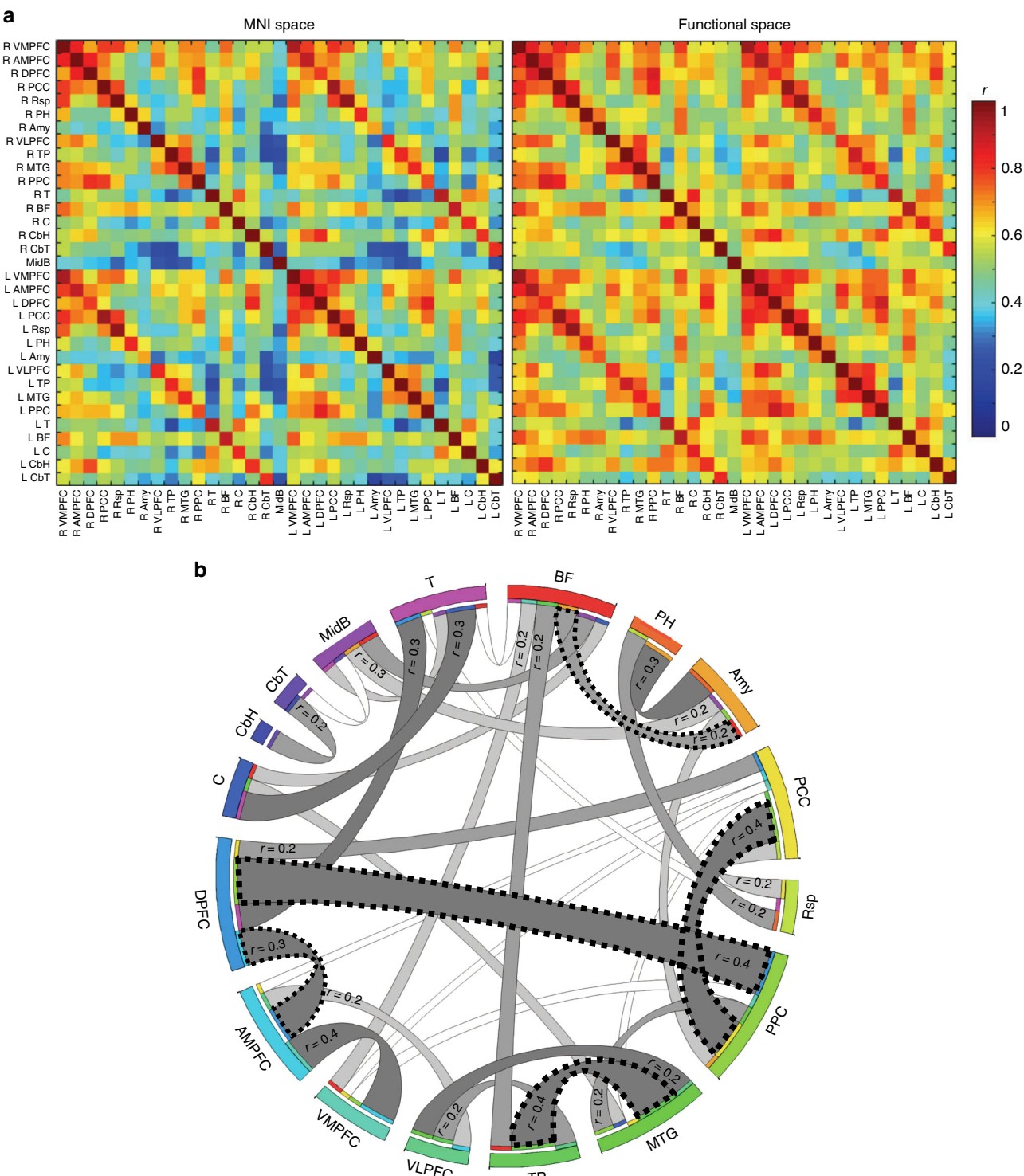

**Fig. 3 Functional connectivity. a** Matrices of the Pearson's correlations between rs-*f*MRI time series of the regions of interest in the structural and in the functional space. **b** Graph representation of the partial correlations between regions of interest in the functional space (connections with partial correlation above 0.2 are depicted; darker grey tones represent stronger connections). Statistically significant partial correlations have a dashed borderline (one-sample *t* test; *p* < 0.0001, which corresponds to the Bonferroni corrected level of significance). The left side structures are not represented, for a clearer visualisation. DPFC dorsal prefrontal cortex, PPC posterior parietal cortex, VLPFC ventrolateral prefrontal cortex, Rsp retrosplenial cortex, MTG middle temporal gyrus, PCC posterior cingulate cortex, C caudate, DPFC dorsal prefrontal cortex, AMPFC antero-median prefrontal cortex, VMPFC ventro-median prefrontal cortex, TP temporal pole, BF basal forebrain, T thalamus, PH parahippocampal region, CbH cerebellar hemisphere, CbT cerebellar tonsil, Amy amygdala, MidB midbrain. *n* = 20 participants

**Table 2 Regions of interest**

| Region of interest | Voxels | MNI (X) | MNI (Y) | MNI (Z) |
|---|---|---|---|---|
| Left ventro-median prefrontal cortex | 2267 | −11 | 55 | −5 |
| Right ventro-median prefrontal cortex | 2673 | 11 | 53 | −6 |
| Left antero-median prefrontal cortex | 2243 | −10 | 50 | 20 |
| Right antero-median prefrontal cortex | 2144 | 10 | 50 | 19 |
| Left dorsal prefrontal cortex | 3818 | −20 | 31 | 46 |
| Right dorsal prefrontal cortex | 3084 | 23 | 32 | 46 |
| Left posterior cingulate cortex | 2484 | −5 | −50 | 35 |
| Right posterior cingulate cortex | 2224 | 7 | −51 | 34 |
| Left retrosplenial cortex | 845 | −6 | −55 | 12 |
| Right retrosplenial cortex | 638 | 6 | −54 | 13 |
| Left posterior parietal cortex | 2448 | −46 | −64 | 33 |
| Right posterior parietal cortex | 1733 | 50 | −59 | 34 |
| Left middle temporal gyrus | 2406 | −58 | −21 | −15 |
| Right middle temporal gyrus | 2170 | 59 | −17 | −18 |
| Left temporal pole | 348 | −38 | 17 | −34 |
| Right temporal pole | 318 | 43 | 15 | −35 |
| Left ventrolateral cortex | 706 | −36 | 23 | −16 |
| Right ventrolateral cortex | 487 | 37 | 25 | −16 |
| Left parahippocampal region | 355 | −24 | −30 | −16 |
| Right parahippocampal region | 287 | 26 | −26 | −18 |
| Left amygdala | 66 | −15 | −9 | −18 |
| Right amygdala | 58 | 17 | −8 | −16 |
| Left caudate | 303 | −11 | 12 | 7 |
| Right caudate | 266 | 13 | 11 | 9 |
| Left cerebellar hemisphere | 906 | −26 | −82 | −33 |
| Right cerebellar hemisphere | 1500 | 29 | −79 | −34 |
| Left cerebellar tonsil | 184 | −6 | −57 | −45 |
| Right cerebellar tonsil | 278 | 8 | −53 | −48 |
| Left thalamus | 382 | −7 | −14 | 8 |
| Right thalamus | 305 | 7 | −11 | 8 |
| Left basal forebrain | 456 | −7 | 12 | −12 |
| Right basal forebrain | 351 | 7 | 9 | −12 |
| Midbrain | 65 | −1 | −22 | −21 |

MNI coordinates represent the centre of gravity of each region

The maps of the DMN registered in the functional space revealed previously underappreciated parts of this network, such as basal forebrain and anterior and mediodorsal thalamic nuclei. Tractography analysis yielded structural connectivity of these new DMN regions to the other regions of the network. Mainly, the cingulum connected the basal forebrain with the medial prefrontal cortex, posterior cingulate region, retrosplenial cortex and hippocampus and parahippocampal regions[59,60]. The fornix linked basal forebrain, specifically the medial septal nuclei, with the hippocampus and parahippocampal regions[49,61,62]. The anterior thalamic projections connected the thalamus with medial and ventrolateral prefrontal regions[63] and finally some of the most medial fibres connecting the basal forebrain with the thalamus, probably corresponded to the mammillothalamic tract of Vicq D'Azyr[64,65]. The graph theory approach[66] applied to the measures of structural connectivity revealed a high node degree of the basal forebrain and the thalamus in the network, as well as high betweenness centrality[67]. These results indicate that basal forebrain and thalamus have high centrality within the network, and therefore can have an important role for network integration and resilience[66–68], along with the classically defined hubs, such as the medial prefrontal region as well as posterior cingulate and retrosplenial region[69].

The involvement of basal forebrain and anterior and medio-dorsal thalamus in the DMN has theoretical and functional repercussions beyond the purely anatomical level. The involvement of the anterior and mediodorsal thalamic nuclei as well as the basal forebrain are concordant with the role of the DMN in memory processes[8,70], as all these regions are relays of the unitary model of the limbic system[49–51]. Previous reports of engagement of the mediodorsal thalamic nucleus and the DMN during memory tasks and the memory deficits provoked by lesions of the anterior and the mediodorsal thalamic nuclei also support this claim[71–74]. At a neurochemical level, the basal forebrain is also a principal actor in the production of acetylcholine[75]. Acetylcholine has a physiological and a neuropharmacological effect on memory processes. For instance, cholinergic system mediates rhythmic oscillation in the hippocampus that facilitates encoding[39,76]. The basal forebrain also contains GABAergic and glutamatergic neurons that mediate hippocampal theta synchronisation through an indirect septo-hippocampal pathway[77]. By providing evidence of the involvement of medial septal cholinergic nucleus and its structural connection to the hippocampus in our DMN model, the present work indicates a match between connectivity, neurochemistry and cognition.

The same correspondence between connectivity, neurochemistry and cognition applies to the relation between DMN and emotional modulation[78–83]. The nucleus accumbens is a central output for the dopaminergic projections and is involved in emotion regulation and affect integration[84,85]. The nucleus accumbens also receives glutamatergic inputs from the hippocampus and the prefrontal cortex[86] belonging to the DMN. Surprisingly, our analysis also revealed the ventral tegmental area, which is also a dopaminergic nucleus with projections to the nucleus accumbens and the medial prefrontal cortex[87]. This association with the mesolimbic dopaminergic pathway is reinforced by our results of functional connectivity, since ventromedial prefrontal cortex and midbrain were among the structures with highest partial correlations with basal forebrain. Hence, combining present and previous findings, we speculate that the DMN, as defined by functional connectivity, might have a putative role in the integration of cholinergic and dopaminergic systems dedicated to memory and emotion.

The new DMN's subcortical structures identified in the current work have cognitive and neurochemical roles that open a new window to the understanding of distinct brain pathologies affecting DMN connectivity (Table 4). Indeed, functional connectivity in each area of the DMN is an estimate of the global coherence of the DMN. Since we demonstrated that limbic thalamus and basal forebrain are nodes with high degree and high centrality in the DMN, damage in these structures should lead to a drastic decrease of functional connectivity in the whole DMN[88]. For instance, Alzheimer's disease is associated with degeneration of the cholinergic system, including the medial septal nuclei, even in the earliest clinical stages of Mild Cognitive Impairment[89] and apparently related to decreased functional coherence and deactivation in hub regions of DMN[90,91]. The high centrality of the basal forebrain in the DMN network may explain this early link between DMN and Alzheimer's disease. In schizophrenia, also associated with decreased DMN connectivity and activation[92,93], neuropathological evidence suggests an abnormal glutamatergic–dopaminergic interaction at the level of nucleus accumbens[94]. In addition, the ventral tegmental area is connected to the nucleus accumbens through the mesolimbic system, the classical dopaminergic pathway associated with schizophrenia, and functional data show a decrease of connectivity between VTA and several brain regions, including the thalamus, in unmedicated schizophrenic patients[95]. The pathophysiology of others diseases, such as drug addiction, depression, temporal lobe epilepsy and

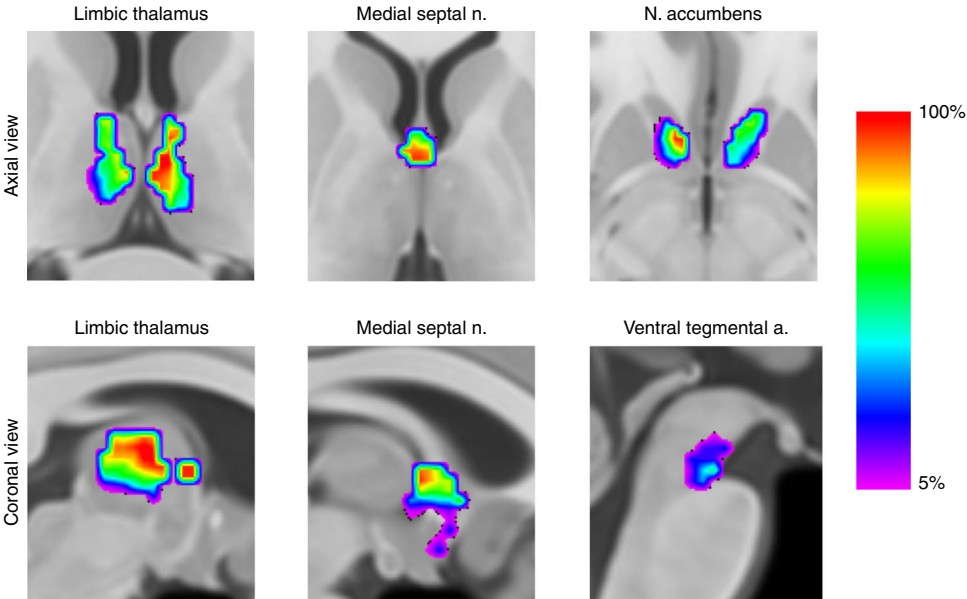

**Fig. 4** Density maps of the functionally aligned individual DMN networks superimposed in the MNI152 space. Colour bar represents the percentage of individuals with a significant correlation in each voxel. This map is freely available at https://neurovault.org/collections/CTTXXAYJ/. n = 20 participants

**Table 3 Proportion of individual DMN maps intersecting, average proportion of the intersection and average volume of the intersection for each nucleus**

| Nuclei | Percentage of individual DMN maps intersecting the nuclei | Percentage of nuclei volume intersected—median [IQR] | Absolute volume of intersection— median [IQR] |
|---|---|---|---|
| L middle septal nuclei | 100% | 22% [15–29] | 248 mm³ [169–327] |
| R middle septal nuclei | 100% | 26% [21–30] | 291 mm³ [235–336] |
| L mediodorsal thalamic | 100% | 64% [53–77] | 696 mm³ [577–838] |
| R mediodorsal thalamic | 100% | 33% [11–42] | 401 mm³ [134–511] |
| L anterior thalamic | 100% | 80% [75–88] | 160 mm³ [150–176] |
| R anterior thalamic | 95% | 62% [28–60] | 193 mm³ [87–187] |
| L nucleus accumbens | 100% | 51% [15–84] | 367 mm³ [108–605] |
| R nucleus accumbens | 95% | 49% [33–58] | 333 mm³ [224–394] |
| Ventral Tegmental Area | 90% | 17% [6–24] | 48 mm³ [17–67] |
| L mammillary body | 50% | 2% [0–9] | 4 mm³ [0–17] |
| L Meynert nucleus | 40% | 0% [0–1] | 0 mm³ [0–22] |
| R Meynert nucleus | 45% | 0% [0–4] | 0 mm³ [0–97] |

*IQR interquartile range, L left, R right*

attention-deficit and hyperactivity disorder involve modifications in the nucleus accumbens, medial septal nuclei or the thalamic nuclei that connect limbic regions as well as dysfunctional connectivity of the DMN (specified in Table 4[27,96–103]). Hence, the involvement of the basal forebrain and the thalamic nuclei in the DMN appears to bridge the gap between the subcortical anatomical differences and the global differences in the DMN previously reported.

Our results is also concordant with recent findings. For instance, Kernbach et al. recently demonstrated that grey matter variability across the DMN can well predict population variation in the microstructural properties of core white matter tracts of the limbic system—the anterior thalamic radiation and the fornix in 10,000 people[9]. As another example, Margulies et al. studied functional gradients along cortical surface and found that DMN areas were at the opposite end of primary motor/sensory areas in a spectrum of connectivity differentiation and that DMN areas exhibit the most considerable geodesic distance at the cortical level, being equidistant to the unimodal cortical areas[11]. These investigators suggested that the DMN acts as a neural relay for transmodal information. We speculate that the thalamus and

basal forebrain may follow the same model at a subcortical level, integrating functional networks related to primary functions and brainstem inputs to the associative areas[104,105].

Interestingly, the thalamus and the basal forebrain are phylogenetically older than many cortical structures, and especially those that compose the DMN[106–108]. The inclusion of these structures in the anatomical model of the DMN can open a window to the exploration of DMN in other mammalian species as well[109–112]. The medial thalamus has already been shown to be part of the mouse's DMN[113], as found in rs-fMRI studies[114–116]. Our results are also concordant with recent neurophysiological evidence in rats about the influence of basal forebrain in the regulation of the DMN[53]. Namely, it has been shown that gamma-band local field potentials in the basal forebrain exerts influence on one of the hub regions of rat's DMN, the anterior cingulate cortex[53]. The cellular basis for the association of these two areas has been characterised with retrograde tracing studies[117,118]. In humans, it has been demonstrated that basal forebrain has a major functional coupling with the anterior cingulate cortex and with ventromedial prefrontal cortex[119]. We speculate that the putative differences in the basal forebrain

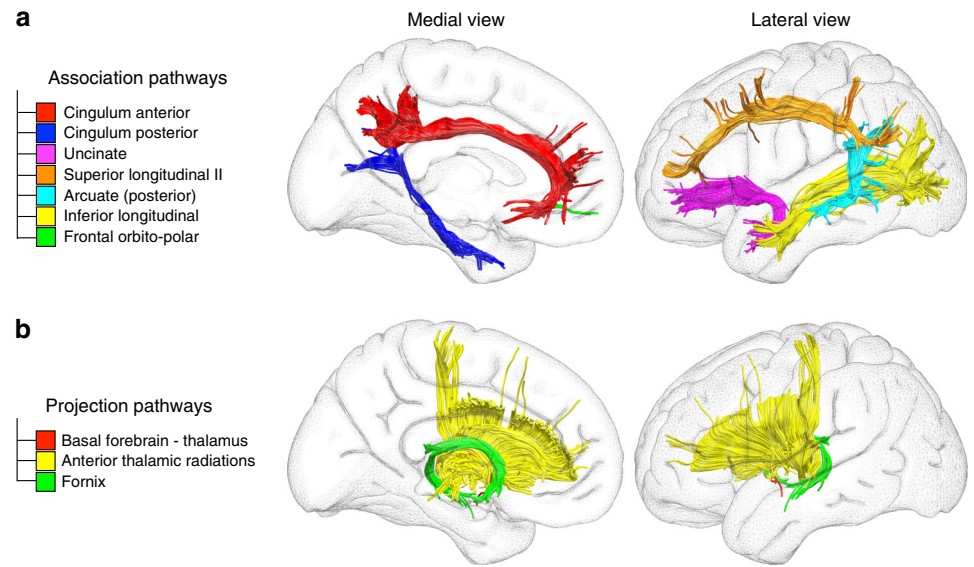

**Fig. 5** Structural connections supporting the DMN. **a** corresponds to the association pathways connecting the cortical regions of the DMN. **b** illustrates the projection pathways mediating the connections between subcortical and cortical regions of the DMN

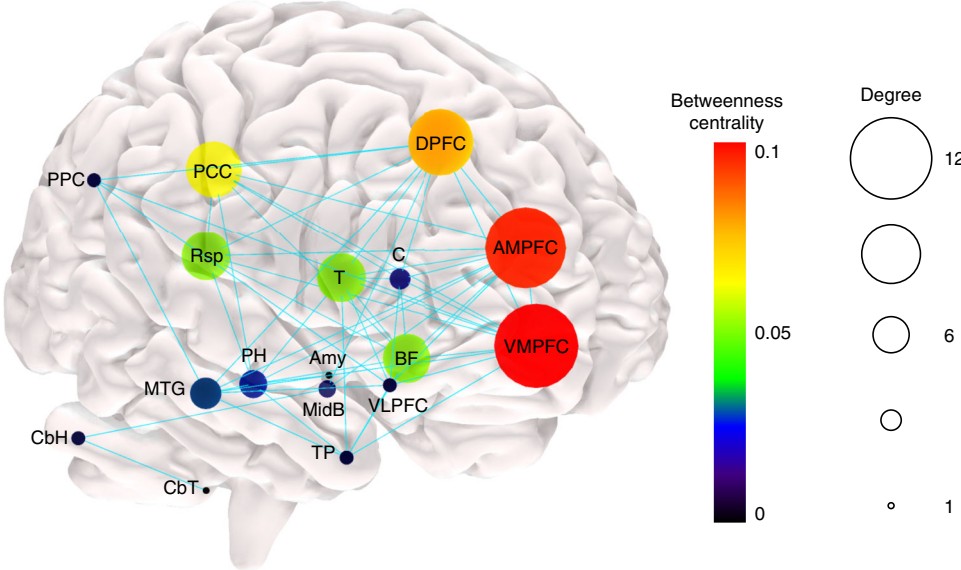

**Fig. 6** Graph theory analysis of structural connectivity. The node size represents node degree and the node colour illustrates node betweenness centrality. The edges denote presence of structural connection. DPFC dorsal prefrontal cortex, PPC posterior parietal cortex, VLPFC ventrolateral prefrontal cortex, Rsp retrosplenial cortex, MTG middle temporal gyrus, PCC posterior cingulate cortex, C caudate, DPFC dorsal prefrontal cortex, AMPFC antero-median prefrontal cortex, VMPFC ventro-median prefrontal cortex, TP temporal pole, BF basal forebrain, T thalamus, PH parahippocampal region, CbH cerebellar hemisphere, CbT cerebellar tonsil, Amy amygdala, MidB midbrain. $n = 20$ participants

projections may be one explanation for the more diffuse DMN activation at the midline level in rats when compared with humans[112].

One limitation of this work is that the overlap between the DMN map and the nuclei studied is based on comparison with templates or variability maps, and not with the individual location of the nucleus in the explored subjects. The limited capacity of structural MRI to differentiate these small nuclei does not allow such comparison. Besides, the reverse transformation of the DMN from the functional space to the MNI space may not be exact, due to inherent limitations of inverse transformations. Although these limitations may decrease the accuracy of the intersection quantification with discrete nuclei, they did not alter the apparent overlap with basal forebrain and with the thalamus.

Furthermore, demonstrating that the additional step of functional alignment results in higher correlation values is somewhat circular. However, our purpose was to demonstrate that this increase in correlation revealed subcortical structures that were previously neglected in the literature. The intra-individual functional connectivity variation across time may also be a source of bias[35]. However, this factor and the lack of uniformity regarding the optimal functional alignment method have not precluded the achievement of higher accurate results in previous studies[43–46]. Finally, tractography analyses can produce inaccurate results[120,121]. In order to avoid these caveats, we employed methods that have previously demonstrated high anatomical reliability when compared with axonal tracing and with post-mortem dissections[59,122–124].

**Table 4 Pathophysiological associations of the subcortical structures of DMN**

| Disease associated with DMN dysfunction | Pathophysiological associations with subcortical DMN | References |
|---|---|---|
| Alzheimer's disease | Degeneration of cholinergic system (including medial septal nuclei), even in MCI stage | 89 |
| | Dysfunction of limbic thalamus in early stages of the disease | 169 |
| | Decreased functional coherence and deactivation in hub regions of DMN | 90,91 |
| Schizophrenia | Abnormalities in glutamatergic–dopaminergic interaction in the nucleus accumbens | 94 |
| | Decreased functional connectivity between the ventral tegmental area and thalamus | 95 |
| | Decreased connectivity and activation of hub regions of DMN | 92,93 |
| TLE | Dysfunctional increase in the hippocampus-mediodorsal thalamus connectivity | 97 |
| | Stimulation of anterior thalamic nucleus is efficacious in treatment of TLE | 170,171 |
| | Enlargement of medial septal nuclei in TLE | 96 |
| | Decreased functional connectivity between the hippocampus and DMN | 27 |
| Depression | Resting activity of the thalamus predicts response to anti-depressant medication | 102 |
| | Nucleus accumbens mediates response to stress and to anti-depressant medication | 100 |

*MCI* mild cognitive impairment, *TLE* temporal lobe epilepsy

In conclusion, this work demonstrates that the registration of individual DMN maps in a functional space improves the definition of the anatomy of DMN by including additional structures, such as the thalamus and basal forebrain. Future research should focus on the cascade of neurochemical and pathophysiological events that follow small subcortical lesions of the DMN.

## Methods

The local ethics committee (Comité de Protection des Personnes "CPP Ile de France V") approved the experiment, and all participants provided written informed consent.

**Subjects and MRI acquisition**. MRI images of subjects without neurological or psychiatric disease were obtained (age mean ± SD 29 ± 6 years, range 22–42 years; 11 females, 9 males) with a Siemens 3 Tesla Prisma system equipped with a 64-channel head coil.

An axial 3D T1-weighted imaging data set covering the whole head was acquired for each participant (286 slices, voxel resolution = 0.7 mm$^3$, echo time (TE) = 2.17 ms, repetition time (TR) = 2400 ms, flip angle = 9°).

rs-*f*MRI images were obtained using T2*-weighted echo-planar imaging (EPI) with blood oxygenation level-dependent (BOLD) contrast. EPIs (TR/TE = 2050/25 ms) comprised 42 axial slices acquired with a multiband pulse[125–128] covering the entire cerebrum (voxel size = 3 mm$^3$) including 290 brain scan volumes in one run of 10 min.

A diffusion-weighted imaging (DWI) acquisition sequence, fully optimised for tractography, provided isotropic (1.7 × 1.7 × 1.7 mm) resolution and coverage of the whole head with a posterior–anterior phase of acquisition, with an echo time (TE) = 75 ms. A repetition time (TR) equivalent to 3500 ms was used. At each slice location, six images were acquired with no diffusion gradient applied (*b*-value of 0 s mm$^{-2}$). In addition, 60 diffusion-weighted images were acquired, in which gradient directions were uniformly distributed on the hemisphere with electrostatic repulsion. The diffusion weighting was equal to a *b*-value of 2000 s mm$^{-2}$. This sequence was fully repeated with reversed phase-encode blips. This provides us with two data sets with distortions going in opposite directions. From these pairs, the susceptibility-induced off-resonance field was estimated using a method similar to that described in ref. [129] and corrected on the whole diffusion-weighted data set using the tool TOPUP and EDDY as implemented in FSL[130].

**rs-*f*MRI analysis—overview**. rs-*f*MRI images were corrected for artefacts in the Funcon-Preprocessing tool of the Brain Connectivity and Behaviour toolkit (http://toolkit.bcblab.com[131–133]). Then, they were registered to T1 high-resolution individual structural images and normalised to MNI152 standard space using Advanced Normalization Tools (ANTS; http://stnava.github.io/ANTs;[134,135]). Preprocessing steps in details:

a. *f*MRI images were first motion corrected using MCFLIRT[132], then corrected for slice timing, smoothed with a full half-width maximum equal to 1.5 times the largest voxel dimension and finally filtered for low temporal frequencies using a Gaussian-weighted local fit to a straight line. These steps are available in FEAT, as part of FSL package[133].

b. *f*MRI images were linearly registered to the T1 images, and subsequently to the MNI152 template (2 mm) using affine and diffeomorphic transformations[136,137]. Confounding signals were discarded from *f*MRI by regressing out a confound matrix from the functional data. The confound matrix included the estimated motion parameters obtained from the previously performed motion correction, the first eigenvariate of the white

matter and cerebrospinal fluid, as well as their first derivative. Eigenvariates were extracted using fslmeans combined with the–eig option. White matter and cerebrospinal fluid eigenvariates were extracted using masks based on the T1-derived 3-classes segmentation thresholded to a probability value of 0.9, registered to the rs-*f*MRI images and binarised. Finally, the first derivative of the motion parameters, white matter and cerebrospinal fluid signal were calculated by linear convolution between their time course and a [−1 0 1] vector.

c. Since the rs-*f*MRI signal can be heavily affected by motion, even following motion correction between temporally adjacent volumes[138], we estimated the signal fluctuation associated with motion and regressed it out from the *f*MRI. To this aim, we employed a recently developed and validated procedure based on data-driven Independent Component Analysis (ICA), termed ICA-Aroma[139]. This method performs an ICA decomposition of the data and estimates which components reflect motion-related noise in the *f*MRI signal on the basis of a robust set of spatial and temporal features. This is made possible due the distinctiveness of the motion-related components isolated by ICA on the fMRI signal[140]. This approach outperforms other methods such as the regression of the motion parameter estimates, while limiting in the same time the loss in degrees of freedom[139]. Compared with spike removal methods such as scrubbing[141], ICA-Aroma has the advantage of preserving the temporal structure of the fMRI signal.

**Individual DMN maps in the structural space**. Individual subject-tailored/fitted DMN maps were obtained by correlation with seed regions of interest of a functional parcellated brain template. The regions used for the seed-based functional connectivity analysis were those defined as DMN regions in the resting-state parcellation map by Gordon et al.[54,142,143]. Gordon et al. created these parcellations according to abrupt changes in resting-state's time course profile, each parcel having a homogeneous time course profile. This general-purpose atlas provides a total of 40 DMN nodes (20 in each hemisphere). The DMN nodes as defined by Gordon et al. are available in the Supplementary Fig. 7 and Supplementary Table 3, respectively). These 40 regions were used as seeds. The correlation map for each seed was obtained using the Funcon-Connectivity tool of the Brain Connectivity and Behaviour toolkit (http://toolkit.bcblab.com;[131]). This tool calculates the Pearson's correlation between the mean rs-*f*MRI time course of the seed with the rest of the brain to generate the functional connectivity map for every seed[131].

The DMN map of each participant was obtained by calculating the median of the 40 seed-based correlation maps of each individual using FSL[144]. We then calculated the median across the DMN maps of the 20 participants to obtain the median DMN map in the MNI152 space. The median was used, instead of the mean, because it is less sensitive to outliers, being a better centrality measure for small sample sizes[145].

**Individual DMN maps in the functional space**. To achieve the proposed optimised map of the DMN, the same 20 individual maps were functionally aligned with each other in a new functional space using the following steps:

Individual DMN maps were aligned with each other using ANTs' script buildtemplateparallel.sh, defining cross-correlation as the similarity measure and greedy SyN as the transformation model[136,137,146]. This approach consists in an iterative (*n* = 4) diffeomorphic transformation to a common space. The group map was obtained by calculating the median of all DMN maps after functional alignment.

The resulting maps correspond to alignment of the 20 individual DMN maps in a functional space (Fig. 2).

**Functional connectivity comparison of the two DMN.** We applied a 0.3 threshold on the functional connectivity strength for both the functional and structural-based DMN maps. This value was chosen because it corresponds to a medium effect size[147]. Time series of rs-*f*MRI of each individual in the different regions of interest identified in the DMN maps were extracted and the mean value of the voxels of each region was obtained, using the command fslstats (with the options –k and –M)[144]. Cortical regions of interest were defined according to the previous anatomical models of the DMN[5,8]. Subcortical regions of interest were defined manually based on the experienced judgement of the neuroanatomists among the authors combined with the careful comparison with previously published atlases[148,149]. Correlation and partial correlation coefficients were determined as measures of functional connectivity.

Two matrices, representing the median correlation values in the MNI152 space and in the functional space, were created using BrainNet Viewer[150]. A paired *t* test was calculated for each cell of the two connectivity matrices using Python's Scipy package, version 0.19.1 (https://www.scipy.org; scipy.stats.ttest_rel). A *p*-value corrected for multiple comparisons at a threshold of <0.0001 (Bonferroni correction) was used. Circos software was used to illustrate functional connections in the functional space (http://circos.ca;[151]).

Two alternative correlation matrices using cortical DMN areas according to Gordon et al.[54], instead of the ones defined in classical anatomical models of Andrews-Hanna et al.[8] and Buckner et al.[5], were calculated.

**Anatomical validation in thalamic, basal forebrain and mesencephalic areas.** Meynert nuclei, medial septal nuclei and diagonal band of Broca probabilistic maps were derived from the work of Zaborszky et al., the Harvard-Oxford probabilistic atlas was employed for the nucleus accumbens, Talairach atlas registered to the MNI152 space for mammillary bodies and thalamic nuclei, and Harvard Ascending Arousal Network Atlas for ventral tegmental area[39,55–58]. A percentage of volume overlap between the DMN map and each nucleus of interest was subsequently calculated for each subject. The same Pearson's correlation threshold of 0.3 was applied for this analysis, as specified in 2.2.3.

**Tractography analysis.** Diffusion-Weighted Images were corrected for signal drift[152], motion and eddy current artefacts using ExploreDTI (http://www.exploredti.com;[153]).

Whole-brain tractography was performed on the software StarTrack using a deterministic approach (https://www.mr-startrack.com). A damped Richardson–Lucy algorithm was applied for spherical deconvolutions[154]. A fixed fibre response corresponding to a shape factor of $\alpha = 1.5 \times 10.3$ mm$^2$/s was adopted. The defined number of iterations was 150, and the geometric damping parameter was 8. The absolute threshold was defined as three times the spherical fibre orientation distribution (FOD) of a grey matter isotropic voxel and the relative threshold as 8% of the maximum amplitude of the FOD[155]. A modified Euler algorithm was used[123]. An angle threshold of 35°, a step size of 0.85 mm and a minimum length of 20 mm were chosen.

Diffusion tensor images were registered to the MNI152 standard space and, then, into the functional space applying the affine and diffeomorphic deformation generated in the previous sections, using the tool tractmath as part of the software package Tract Querier[156].

The same regions of interest used for functional connectivity calculation (section 2.2.3) were also used for the tractography analysis.

The command tckedit of MRtrix toolbox (http://www.mrtrix.org/[157]) was used to extract the tracts of interest. The –include arguments were composed by combinations of DMN regions of interest to guarantee that the selected streamlines crossed at least two DMN regions of interest.

In order to have a group-representative map for each tract, individual tracts were converted into maps using tckmap command of MRtrix toolbox (http://www.mrtrix.org;[158]). Each voxel was binarised in 1 or 0 according to being or not being intersected by a streamline[159]. Lastly, a one-sample *t* test was calculated using FSL randomise, with variance smoothing of 4 mm[159]. The resulting group-representative tract maps using this method were shown to have a good anatomical correspondence with histological atlas of white matter tracts[159,160]. BrainVisa was used to create the corresponding illustration of the tracts that reached significance[161].

**Graph theory analysis of structural connectivity.** To explore whether the new putative regions of the DMN are essential structures and potential areas of vulnerability in the network, we investigated the hub properties of the network nodes using graph theory measures, namely node degree and betweenness centrality[66]. Node degree refers to the number of connections between a given node and the other nodes of the network. Betweenness centrality is the fraction of all shortest paths in the network that pass through a given node[66].

For each participant, an anatomical connectome matrix of the DMN was built using the tck2connectome command of MRtrix (http://www.mrtrix.org;[157]). Each region of interest was defined as a node. Only the streamlines that ended in both regions of interest were considered[162,163]. The matrices were binarised depending on the existence or absence of streamlines connecting two regions of interest[162,163]. The defined threshold for the binarisation was 1 because the number of streamlines

does not reflect the connectivity strength or the true number of axonal projections between two brain regions[163,164], and previous evidence has shown that changing the streamline count threshold for binarisation (between 1 and 5) does not change the overall results of the network analysis. Brain Connectivity Toolbox for Python (https://pypi.python.org/pypi/bctpy) was used to obtain the network measures. The functions degrees_und and betweenness_bin were run, respectively, for each individual matrix[67]. The median values for each measure were obtained. The illustration of the network was made using Surf Ice (https://www.nitrc.org/projects/surfice/).

**Statistics and reproducibility.** In order to assess reproducibility, the effects of interest were also measured at the individual level (i.e., replicated for every subject).

**Reporting summary.** Further information on research design is available in the Nature Research Reporting Summary linked to this article.

## Data availability

The raw data used for this research are available on demand from the corresponding authors pedroascimentoalves@gmail.com and michel.thiebaut@gmail.com. The data for the statistical comparisons presented in the paper are available in Supplementary Data 1–5. Unthresholded statistical maps of the comparison between the two methods of alignment—structurally and functionally aligned DMN—are available at neurovault.org (https://neurovault.org/collections/OCAMCQFK/).

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

## Acknowledgements

P.N.A. work was financially supported by the Research Experience Fellowship grant of the European Academy of Neurology and by "PRÉMIO JOÃO LOBO ANTUNES" - SCML. The research leading to these results received funding from the "Agence Nationale de la Recherche" [grant number ANR-13- JSV4-0001-01]. This project has received funding from the European Research Council (ERC) under the European Union's Horizon 2020 research and innovation programme (grant agreement No. 818521) and from the Fondation pour la Recherche Médicale (FRM DEQ20150331725).

## Author contributions

P.N.A. implemented part of the methods, performed the analyses and wrote the manuscript. C.F. implemented part of the methods and edited the paper, Vyacheslav Karolis contributed to the methods and edited the paper, D.B., D.S.M. and E.V. helped conceptually and edited the paper, M.T.S. conceived and coordinated the study, reviewed the neuroimaging data, wrote the paper and provided funding.

## Competing interests

Michel Thiebaut de Schotten is an Editorial Board Member for *Communications Biology*, but was neither involved in the editorial review of nor the decision to publish this article. The authors declare no other competing financial or non-financial interests.
