## [Peer Review File · Communications Biology]

Reviewers' comments:

Reviewer #1 (Remarks to the Author):

In this well-written paper titled "Subcortical Anatomy of the Default Mode Network: a functional and structural connectivity study", the authors propose a neuro-anatomical model of the Default Mode Network (DMN) that includes subcortical structures. Using volumetric functional alignment, they were able to show that additional structures belong to the DMN, including the thalamus and the basal forebrain. Moreover, they provide a qualitative and quantitative description of the importance of these structures in the DMN, using tractography.

This is a timely contribution to the investigation of the DMN and I believe it should be of high interest for the community. The methodology used is extremely relevant and it might inspire a lot of researchers interested in functional connectivity.

I have one major and several minor comments.

Major:

The definition of regions of interest was globally lacking transparency and/or clarity:

- o For the DMN individual maps, the authors cited Gordon et al 2016 to justify the selection of seeds. By quickly having a look at this paper, it is not obvious for me what are the 20 seeds. The author should provide a map (or a table at least) to show them.

- o In Figure 2e, we see the 3 "hypothesized" subcortical regions. For consistency across figures, the authors could represent those three structures in a more complete diagram in Figure 1c.

- o The selection of region of interest for the functional connectivity comparison is not explained. I believe it has been done in the same way as the description lines 250-255 in the tractography section. However, this last description is not clear: are they defined according to the literature (Andrews-Hanna et al., 2010; Buckner et al., 2008) or based on the results? ("The map of the DMN created through functional alignment was used to define the regions of interest"). None of this selection is a problem for the tractography analysis but the latter might be a problem for the functional connectivity comparison. Indeed, if we select ROI based on the DMN aligned map, then it seems logical to find higher correlations in the functional space. If the ROI selection had been done in the structural space, isn't it exactly the opposite result that we would observe? Except if I miss something here I am afraid it could be some double dipping. Could the authors show the same analysis but with ROI based in the structural space? Or explain how it is not double dipping? Why not taking the 40 seeds of interest from Gordon et al 2016?

- o For the ROI defined according to the DMN functionally aligned map, with which statistical threshold? Line 329, it is not clear how the 9 additional regions are defined (regions that are found in the functionally aligned but not on the structurally aligned?).

Minor:

1. In the introduction, the authors could mention Glasser et al (2016), in which a new method of brain data alignment is proposed, based on multiple brain measures such as functional connectivity, sulcal depth of topography. This study focuses on cortical areas (and on surfaces) but has proven how efficient a functional alignment could be to investigate brain networks. I think it supports even more

the need for a similar work on subcortical regions.

2. While the authors mention in the discussion (lines 483-486) why an anatomical alignment of subcortical regions is difficult, I think it should also be mentioned in the introduction, in order to support the rationale for aligning subcortical regions on function.

3. A study by Cunningham et al (2016), investigating the link between the DMN and the thalamus, could be cited (<https://doi.org/10.1002/hbm.23429>) and discussed.

4. Statistics in the tractography results are not reported.

5. Typos:

- figure 2e legend: FWE corrected instead of FEW corrected.

- line 297, should be figure 2e?

- figure 3a right: functional space, figure 3b DPFC and PCC: problem with the font

Reviewer #2 (Remarks to the Author):

The study provides human fMRI evidence linking the basal forebrain and thalamic nuclei to the DMN. This is a finding of substantial interest and supports recent animal literature in this area. This is an important contribution to the literature, but I have several concerns about some details of the study as outlined below.

The motivation for the study as well as the previous relevant literature are well described in the introduction.

3.1: The authors state that basal forebrain and thalamic nuclei are seen only in the functional aligned map and not in the MNI52 map. This is an important finding, and it seems important to provide statistics related to this observed difference, at present it is only stated that a significant difference was observed. Also, I would like to see the extent of the activation in individual subjects comparing the two methods, so we can get an idea of inter-subject variability.

3.2: The title of the correlation maps are illegible as are some legends in Fig.3b. For the BF, are there any areas for which the partial correlations are weaker in the functional method? It would be useful to see a more complete list of the partial correlations for BF in the functional method, maybe in form of a table. How robust are these results in individual subjects?

3.3 The colormap is hard to read, because both low and high values are red. Please modify. It would also be important to show some of the areas with low inter individual overlap; right now only high overlap regions are shown.

3.4 Tract tracing (DTI) is generally difficult to interpret especially if fibers are part of large bundles, such that 1:1 connectivity between source and target cannot be guaranteed. The authors should mention limitations of DTI; personally I do not think this part adds much to the study.

3.5 The graph theory part of the analyses is of interest and should be expanded. The corresponding figure is illegible, please correct. The path length/connectedness parameters should be shown more clearly for different structures (and in individual subjects), to convince readers of the high centrality of thalamus and BF DMN nodes.

p.12: It would be useful to highlight briefly other studies that have used functional alignment, and discuss the limitations / advantages of this method. Two or three specific examples from other systems neuroscience areas would be useful for illustration.

p.14: It should be mentioned that the BF also contains GABAergic and glutamatergic populations of neurons with diverse projection patterns in addition to the cholinergic cells.

p.15: The considerations of the role of DMN in various disorders is a little too superficial to be useful. Can the authors focus on the putative specific role of the DMN, for example hypo- or hyperactivation in specific disorders mentioned?

p.15: I would also encourage the authors to speculate why BF functional activations (gamma band LFP) are strongly coupled to the medial frontal cortex (i.e. cingulate) in the rat? It seems like the Cingulate is not a major target of the projection in humans? What is the most likely avenue for this BF-ACC coupling given the present data? Some thoughts on this would be appreciated.

Reviewer #3 (Remarks to the Author):

Title: Subcortical Anatomy of the Default Mode Network: a functional and structural connectivity study

GENERAL COMMENTS

The authors present a study in which they attempt to demonstrate the feasibility of including several subcortical areas -- including basal forebrain, anterior and mediodorsal thalamic nuclei, and cholinergic nuclei -- in the definition of the "default mode network" (DMN). They do this on the basis of common functional covariance and deterministic DTI tractography observed in these brain regions.

In general, the approach is a novel attempt to improve our understanding of DMN function, by including subcortical regions that have traditionally not been considered in such analyses. The move to areas beyond the neocortex is a welcome one. It also appears promising to use functional rather than structural coregistration to improve the precision with which the subcortical regions of interest can be localized (but these conclusions have issues; see below). Overall, this study is a nice idea and with some work could be an important addition to the scientific discussion about what the DMN is, and the neural/mental processes it subserves.

I have some criticisms of the approaches used, however, which are detailed below. These should be addressed either by clarification of the methods, re-analysis of the data, or a discussion of the potential pitfalls of the methods where the authors are not inclined to change them (in particular, I am not convinced by the DTI approach).

Additionally, a number of important details are missing from the Methods section, which make it difficult to either reproduce the results reported, properly interpret them, or assess the degree to which they may be accounted for by confounds or statistical artifacts. Specific comments follow.

SPECIFIC COMMENTS

METHODS [line numbers in square brackets]

1. [178] "Resting state functional MRI images were corrected for artefacts" - Which artifacts, what steps were followed? You need to provide detail here, it's not sufficient to say you passed data through a pipeline, without specifying which steps were taken. It is especially important to explain how you dealt with motion correction. Was a global signal regression done? A censoring approach? Removal of temporal derivatives? Because you use functional data to co-align subjects and then find increased functional correlations in the same sample, how do we know this improvement is not due to motion artifacts, which have been demonstrated to produce systematic spurious correlations and changes to apparent topological organization (e.g., Power et al., Neuroimg, 2012).
2. [180] Was normalization linear or also nonlinear?
3. Section 2.2.1. is poorly written. It could be rewritten to provide a step-by-step description of how DMN distributions were computed. Also: how large were the ROIs used as seeds? Was an average time series used? Were individual voxels used? What was correlated with what? Were Pearson correlations used? Regression coefficients? Partial correlations? More detail is required to reproduce this study. Why are medians used instead of means?
4. [197] "To obtain a group DMN map in the structural space, the median of the twenty individual maps was derived." -- Not sure what this means. What is "the structural space"? It becomes clearer later in the text, but you should introduce these terms early and not assume the reader knows what they mean. Also should be more clear that this is a "group median map".
5. [213] "different regions of interest identified in the DMN maps" -- how were these ROIs derived? Was thresholding applied, and if so what was the rationale? Were time-series averaged or was each voxel correlated separately?
6. [220] "p-value corrected for multiple comparisons at a threshold of <0.0001 (Bonferroni correction)." -- Does this mean there were 500 comparisons?
7. [231] "A percentage of volume overlap between the DMN map and each nucleus of interest" -- Were the volumes comparable? What is this quantity meant to show? How dependent is this quantity on the threshold used to define the DMN map (which hasn't been specified)? The rationale for thresholding becomes even more important since you are using these values to draw inferences...
8. [237] Why use a deterministic tractography approach instead of probabilistic? How do you account for variability in streamline distribution? Does this method account for crossing fibres or different anisotropy profiles generally?
9. [250] "Regions of interest were defined according to the previous anatomical models of the DMN" -- It should be explained how these were derived, what their granularity is (and how many there are), why they were chosen over other alternatives, etc. More detail on these ROIs is required as they are a critical aspect of this method.
10. [253] "The command tckedit of MRtrix toolbox (<http://www.mrtrix.org/>; Tournier et al., 2012) was used to isolate the tracts of interest." -- What does it mean to "isolate the tracts of interest"? How does tckedit do this?

11. [256] "In order to have a group-representative tract volume for each tract, one sample T-test of individual tract volumes was calculated using FSL randomise" -- How are "tract volumes" computed? What does this computation depend on? What does it represent in biological terms?

12. [271] "The matrices were binarised (no threshold) for structural connection or no structural connection between regions." -- What does "no threshold" mean? On what basis was this binarization performed? Since you are not using probabilistic tractography, and have no estimate of uncertainty how are you confident this is the "correct" connectome (i.e., no false negatives)? Particularly important since you only have 20 subjects. Would adding more subjects change this connectome (and its topological characterization)?

RESULTS

13. Fig 2d: Higher FC for functionally aligned maps: You use the same subjects to functionally co-align as to compute FC. FC is used as the feature for coalignment. Doesn't it seem trivial (i.e., double-dipping) then that you find stronger FC for the functional versus structural maps? It would be more convincing had you shown this for an independent sample. This should be discussed.

14. [305] "statistically significant in 95% and in 18% of pairs, whether or not correcting and correcting for multiple comparisons was applied" -- If you correct for multiple comparisons (which you should), there's no need to report uncorrected statistics. [Also, the sentence is poorly constructed].

15. [306] "Table 2 represent the MNI coordinates of all the regions of interest in the DMN map" -- Should specify that these coordinates are the center of mass of the ROI.

16. Fig 3b is confusing because it is not clear whether you are showing partial correlation coefficients or differences in these coefficients between structural and functional maps. The significance levels appear to refer to the latter, but the caption indicates these are coefficients only for the functional map. Why use asterisks, instead of changing the edge colours to indicate significance? Why show uncorrected significance, when you clearly need to apply a FWE correction here? Also, the labels are not all readable.

17. Fig 4: The colour map here has red/orange on both ends of the scale, which makes interpretation difficult. This should be fixed to have unique (preferably hot/cool) colours for each end of the scale. Also, units would make more sense as percentages, especially since these are discussed in the corresponding text.

18. The percent overlap metric relies on a threshold, and how this threshold was determined is unclear (statistical significance at 0.05? corrected/uncorrected? FWE? Cluster-level?). Instead of a (necessarily) arbitrary threshold approach, why not simply compute the mean/std correlation values for these voxels? Surely this would highlight the overlapping areas in a less binarized fashion...?

19. [329] "nine additional regions of interest were defined including..." -- how were these defined? Was a threshold set? These details are important!

DISCUSSION

20. [447] "Hence, integrating present and previous findings, it appears that the DMN, as defined by functional connectivity, is at the interplay between a cholinergic and a dopaminergic system dedicated to memory and emotion." -- It is not clear how the present findings elucidate this system, and

neurotransmitter systems in particular, beyond what we already knew. The methods used here do not allow the level of resolution (temporal or spatial) to permit any sort of direct inference with respect to these systems. It is not clear what "at the interplay" refers to. I'd suggest removing these speculative statements, or make it clearer that they are purely speculative and not in any way supported by the evidence presented in this study. It is important to put your work in context, but equally important to avoid unwarranted conclusions.

Reviewer 1:

A) The definition of regions of interest was globally lacking transparency and/or clarity. For the DMN individual maps, the authors cited Gordon et al 2016 to justify the selection of seeds. By quickly having a look at this paper, it is not obvious for me what are the 20 seeds. The author should provide a map (or a table at least) to show them.

R: We thank the reviewer for raising this issue. To clarify what were the regions that were used as seeds, we created a surface map with the cortical regions belonging to the default-mode network according to Gordon et al 2016. A table was also created with the MNI coordinates of the centres of gravity of those regions. These materials were now added to the Supplementary Materials. In addition, the following sentence was added to the manuscript: *“The DMN nodes as defined by Gordon et al. are available in the Supplementary Materials (Figure S1 and Table S1, respectively).”*

This choice was made according to previous works in the literature. The following references were added:

- Shine, J.M., Bissett, P.G., Bell, P.T., Koyejo, O., Balsters, J.H., Gorgolewski, K.J., Moodie, C.A., Poldrack, R.A., 2016. The dynamics of functional brain networks: integrated network states during cognitive task performance. *Neuron* 92, 544–554. <https://doi.org/10.1016/j.neuron.2016.09.018>.

- Tomasi, D., Volkow, N.D., 2018. Association between brain activation and functional connectivity. *Cereb Cortex* 1–13. <https://doi.org/10.1093/cercor/bhy077>.

B) In Figure 2e, we see the 3 “hypothesized” subcortical regions. For consistency across figures, the authors could represent those three structures in a more complete diagram in Figure 1c.

R: The diagram of the Figure 1c has now been revised to be consistent with the hypothesized anatomical model. Specifically, the septal nuclei and the nucleus accumbens (basal forebrain nuclei) were added and connected with the appropriate structures according to Aggleton, (2008), Haber et al. (1995, 1990), Kunishio and Haber (1994) and MacLean (1949). The references of the Figure were also updated accordingly.

C) The selection of region of interest for the functional connectivity comparison is not explained. I believe it has been done in the same way as the description lines 250-255 in the tractography section. However, this last description is not clear: are they defined according to the literature (Andrews-Hanna et al., 2010; Buckner et al., 2008) or based on the results? (“The map of the DMN created through functional alignment was used to define the regions of interest”). None of this selection is a problem for the

tractography analysis but the latter might be a problem for the functional connectivity comparison. Indeed, if we select ROI based on the DMN aligned map, then it seems logical to find higher correlations in the functional space. If the ROI selection had been done in the structural space, isn't it exactly the opposite result that we would observe? Except if I miss something here I am afraid it could be some double dipping. Could the authors show the same analysis but with ROI based in the structural space? Or explain how it is not double dipping? Why not taking the 40 seeds of interest from Gordon et al 2016?

R: The definition of the cortical regions of interest for the tractography was made according to established literature (Buckner et al., 2008; Andrews-Hanna et al., 2010). However, previous anatomical models do not include subcortical structures. Therefore, these regions were drawn manually based on the experienced judgment of the neuroanatomists among the authors combined with the careful comparison with previously published atlases (Nieuwenhuys et al. 2008; Catani and Thiebaut de Schotten 2012). A map with these ROIs was now made available in the Supplementary Materials (Figure S2). The same ROIs were used for the functional connectivity portion of the analysis, but we acknowledge that this was unclear in our prior manuscript. To clarify this aspect, the above-mentioned paragraph was reformulated and moved to the functional connectivity section (section 2.2.3): *“Cortical regions of interest were defined according to previous anatomical models of the DMN (Andrews-Hanna et al., 2010; Buckner et al., 2008). Subcortical regions of interest were defined manually based on the experienced judgment of the neuroanatomists among the authors combined with the careful comparison with previously published atlases (Nieuwenhuys et al. 2008; Catani and Thiebaut de Schotten 2012).”* In the tractography section (section 2.3), the following sentence was added: *“The same regions of interest used for functional connectivity (section 2.2.3) were also used for the tractography analysis”*. The use of the same regions of interest for structural and functional connectivity ensures the homogeneity of the analysis.

The Gordon et al. DMN parcellation is a valid alternative to the method we used. As requested by the reviewer, we have repeated the analysis using Gordon et al. areas as the cortical regions of interest. The results are presented in the Supplementary Materials – Figure S4. Pearson's correlation was higher in the functional space in 92% of cells of the correlation matrix, and the difference was statistically significant in 5% of cells (p -value < 0.00004, after Bonferroni's correction for multiple comparisons). Accordingly, the following sentences were added to the methods and to the results section respectively: *“Two alternative correlation matrices using cortical DMN areas according to Gordon et al. (2016), instead of the ones defined in classical anatomical models of Andrews-Hanna et al. (2010) and Buckner et al. (2008), were calculated.”*; *“The alternative correlation matrices using cortical DMN areas*

according to Gordon et al. (2016) are presented in the Supplementary Materials (Figure S4).”

We agree that demonstrating stronger functional connectivity after performing functional alignment may bear some aspect of circularity. However, we originally performed this analysis to demonstrate that the method of functional alignment was valid at the subcortical level. We have now clarified this point in the limitations section of the discussion: *“Furthermore, demonstrating that the additional step of functional alignment results in higher correlation values is somewhat circular. However, our purpose was to demonstrate that this increase in correlation revealed subcortical structures that were previously neglected in the literature.”*

D) For the ROI defined according to the DMN functionally aligned map, with which statistical threshold? Line 329, it is not clear how the 9 additional regions are defined (regions that are found in the functionally aligned but not on the structurally aligned?).

R: We set the statistical threshold at a Pearson correlation of $r=0.3$ to define the ROIs as it corresponds to a medium effect size (Cohen, 1992). This point was clarified in the methods: *“Functional and structural-based DMN maps were thresholded at $r=0.3$ which, corresponds to a medium effect size (Cohen, 1992).”*

Regarding the nine additional regions, they were defined as mentioned in the commentary. We now made this aspect explicit in the manuscript: *“Nine additional regions were defined as being solely revealed after the functional alignment. These regions included the left and right thalamus, the left and right basal forebrain, the midbrain, the left and right ventral lateral prefrontal cortex (VLPFC) and the left and right caudate nucleus (inferior regions of the nuclei), resulting in a total of 33 regions of interest.”*

E) In the introduction, the authors could mention Glasser et al (2016), in which a new method of brain data alignment is proposed, based on multiple brain measures such as functional connectivity, sulcal depth of topography. This study focuses on cortical areas (and on surfaces) but has proven how efficient a functional alignment could be to investigate brain networks. I think it supports even more the need for a similar work on subcortical regions.

R: We thank the reviewer for suggesting this reference, which is now cited in the introduction (when referring to functional alignment) as follows: *“... and have recently enabled much improved models of cortical parcellation (Glasser et al. 2016)”*

F) While the authors mention in the discussion (lines 483-486) why an anatomical alignment of subcortical regions is difficult, I think it should also be mentioned in the

introduction, in order to support the rationale for aligning subcortical regions on function.

R: We appreciate the reviewer's suggestion. To reinforce the rationale at the outset we have now reformulated part of the introduction as follows: *"To address several of these challenges that may impede proper group analysis if relying exclusively on MRI structural landmarks, we propose to revisit the anatomical scaffold of the DMN using a coregistration based on functional alignment."*

G) A study by Cunningham et al (2016), investigating the link between the DMN and the thalamus, could be cited (<https://doi.org/10.1002/hbm.23429>) and discussed.

R: We now added this reference to the manuscript. *"The thalamus has also been shown to be structurally and functionally connected to DMN regions (Cunningham et al., 2016; Fransson, 2005)."*

H) Statistics in the tractography results are not reported.

R: The group representative tracts presented in Figure 5 resulted from the application of one-sample T-test on individual tracts. To report in more detail these results, statistical maps of this calculation were added in the Supplementary Materials – Figure S7. The following sentence was included in the manuscript: *"The statistical maps of the tractography analysis are presented in the Supplementary Materials (Figure S7)."*

I) Typos:

- **Figure 2e legend: FWE corrected instead of FEW corrected.**
- **line 297, should be Figure 2e?**
- **Figure 3a right: functional space, Figure 3b DPFC and PCC: problem with the font.**

R: Typos were corrected. Thank you.

Reviewer 2:

A) 3.1: The authors state that basal forebrain and thalamic nuclei are seen only in the functional aligned map and not in the MNI52 map. This is an important finding, and it seems important to provide statistics related to this observed difference, at present it is only stated that a significant difference was observed. Also, I would like to see the extent of the activation in individual subjects comparing the two methods, so we can get an idea of inter-subject variability.

R: With Figure 2e, we intended to show the results of the statistical comparison between the two methods of alignment within the hypothesized areas. The results are presented at two

levels of significance - family wise error corrected p-values of 0.05 and 0.01. In addition, we have made the unthresholded statistical maps of this comparison openly available in neurovault.org (<https://neurovault.org/collections/OCAMCQFK/>). To clarify the meaning of the figure, its caption was changed to: “(e) statistical comparison (paired t-test) between the two methods of alignment - structurally and functionally aligned DMN — in the three hypothesized regions, with colours indicating statistically significant differences at two levels of significance - <0.05 and <0.01 , family-wise error (FWE) corrected p-values (higher in the functional space).” The following sentence in the text was also modified: “Unthresholded statistical maps of the comparison between the two methods of alignment — structurally and functionally aligned DMN — are available at neurovault.org (<https://neurovault.org/collections/OCAMCQFK/>).”

Regarding the extent of the activation in individual subjects and inter-subject variability, Figure 4 indicates the proportion of participants’ DMN overlapping with the hypothesized regions. This map was made available at <https://neurovault.org/collections/CTTXXAYJ/>. We also clarified the legend of Table 3: “Proportion of individual DMN maps intersecting, average proportion of the intersection and average volume of the intersection for each nucleus.”

To better illustrate the difference of individual DMN maps between the MNI and functional space, two representative cases were presented in the Supplementary Materials. In these examples, we can observe that functional alignment does not cause large deformations on individual maps, but rather induces slight modifications that may overcome the problems of strict anatomical alignment in group-level analysis of functional networks at the subcortical level. The following sentence was added to the manuscript: “Two representative cases of individual maps in the MNI152 space and in the functional space are presented in the Supplementary Materials (Figure S3).”

B) 3.2: The title of the correlation maps are illegible as are some legends in Fig.3b. For the BF, are there any areas for which the partial correlations are weaker in the functional method? It would be useful to see a more complete list of the partial correlations for BF in the functional method, maybe in form of a table. How robust are these results in individual subjects?

R: The titles of Figure 3b were enlarged.

Concerning the partial correlations of the basal forebrain, we did not find any areas with statistically significant stronger or weaker connectivity. Our sample may not have enough power to detect differences. Tables with the median, and with the range and interquartile range of the partial correlation between all the areas is now available in the Supplementary Materials – Tables S3 and S4. In these tables, the partial correlations for the basal forebrain

and for the thalamus were highlighted. In addition, we included a table with the p-values of the comparison of partial correlation between the anatomical and functional alignment method (Supplementary Materials – Table S5).

Regarding Pearson's correlations, the left basal forebrain had significantly higher correlations with the right antero-median prefrontal cortex, with the right posterior parietal cortex and with the midbrain area, while the right basal forebrain had significantly higher correlations with the right temporal pole and with the left cerebellar hemisphere. A table with the p-values of the comparison of Pearson's correlations between the two methods of alignments was also included in the Supplementary Materials (Table S2).

To show the correspondence of these results at individual level, we added a table with the partial correlations in two representative participants (Supplementary Materials Table S6).

The following text was added to the manuscript: *“Regarding the hypothesized areas, the left basal forebrain had significantly higher correlations with the right antero-median prefrontal cortex, with the right posterior parietal cortex and with the midbrain area, while the right basal forebrain had significantly higher correlations with the right temporal pole and with the left cerebellar hemisphere (Supplementary Materials - Table S2). The left thalamus had significantly higher correlations with the left parahippocampal region, with the left temporal pole, with the right and the left antero-median prefrontal cortex, with the left and the right ventro-lateral prefrontal cortex and with the right posterior parietal cortex, while the right thalamus had significantly higher correlations with the right and the left antero-median prefrontal cortex, with the left temporal pole and with the left cerebellar tonsil (Supplementary Materials, Table S2). No significant difference was found for partial correlations. Tables presenting the median, the range and the interquartile range of the partial correlations between all regions of interests, the statistical comparison of partial correlations between the two methods of alignment and the partial correlations of two representative cases are available in the Supplementary Materials (Tables S3, S4, S5 and S6, respectively).”*

C) 3.3 The colormap is hard to read, because both low and high values are red. Please modify. It would also be important to show some of the areas with low inter individual overlap; right now only high overlap regions are shown.

R: The colourmap was changed accordingly. The areas with the lowest interindividual overlap are now more clearly seen, since they have been changed to violet/blue, following the reviewer's advice. The areas with the highest interindividual overlap were kept red.

D) 3.4 Tract tracing (DTI) is generally difficult to interpret especially if fibers are part of large bundles, such that 1:1 connectivity between source and target cannot be

guaranteed. The authors should mention limitations of DTI; personally I do not think this part adds much to the study.

R: We agree that tractography of diffusion weighted imaging has certain limitations.

To explicitly communicate important limitations of the tractography analysis to the reader, two references were added to the manuscript: “Jones, D.K., Cercignani, M., 2010. Twenty-five pitfalls in the analysis of diffusion MRI Data. *NMR Biomed* 23, 803–820” and “Maier-Hein, K.H., Neher, P.F., Descoteaux, M., 2017. The challenge of mapping the human connectome based on diffusion tractography. *Nat Commun* 8, 1349–136”. In addition, the following sentences were included: “*Finally, tractography analyses can produce inaccurate results (Jones and Cercignani, 2010; Maier-Hein et al., 2017). To avoid these caveats, we employed methods that have previously demonstrated high anatomical reliability when compared to axonal tracing and with post-mortem dissections (Catani et al., 2012; Dell’Acqua et al., 2013; Karolis et al., 2019; Thiebaut de Schotten et al., 2011)*”.

E) 3.5 The graph theory part of the analyses is of interest and should be expanded. The corresponding Figure is illegible, please correct. The path length/connectedness parameters should be shown more clearly for different structures (and in individual subjects), to convince readers of the high centrality of thalamus and BF DMN nodes.

R: The font of the legends of the nodes in the graph theory Figure (Figure 6) were substantially increased. We have now also included the table with node degrees and node betweenness centrality into the main manuscript (Table 4). In addition, this table was appended with the interquartile range of node degrees and betweenness centrality of each node in order to provide a measure of the distribution between individual subjects.

Individual examples were also included in the Supplementary Materials (Table S7) with graph theory measures of subject 1 and 2 (the same subjects that were referred in reviewer’s commentaries in A and B).

F) p.12: It would be useful to highlight briefly other studies that have used functional alignment, and discuss the limitations / advantages of this method. Two or three specific examples from other systems neuroscience areas would be useful for illustration.

R: Two new references on functional alignment were added: “Robinson, E.C., et al, 2014. A new flexible framework for Multimodal Surface Matching. *Neuroimage* 100, 414–426.” and “Glasser, M.F., et al. 2016. A multi-modal parcellation of human cerebral cortex. *Nature* 536, 171–178”. The following sentence was added: “... *also have led to better predictions of task activation patterns in group-analysis when compared to morphological alignment methods and have improved models of cortical parcellation (Glasser et al., 2016; Langs et al., 2015)*.” Additionally, to state the limitations of functional alignment, the following text

was included: “*The intra-individual functional connectivity variation across time may also be a source of bias (Braga and Buckner, 2017). However, this factor and the lack of uniformity regarding the optimal functional alignment method have not precluded the achievement of higher accurate results in previous studies (Glasser et al., 2016; Langs et al., 2015; Mueller et al., 2013; Robinson et al., 2014).*”

We appreciate the suggestion to mention similar approaches from other fields of neuroscience. The following sentences and references were added regarding the consistency of our results with the mouse model: “*For instance, the medial thalamus is part of mouse’s DMN (Gozzi and Schwarz, 2016), as found in rs-fMRI studies (Bertero et al., 2018; Liska et al., 2015; Sforazzini et al., 2014). Our results are also concordant with recent neurophysiological evidence in rats about the influence of basal forebrain in the regulation of the DMN (Nair et al., 2018). Namely, it has been shown that gamma-band local field potentials in the basal forebrain exerts influence on one of the hub regions of rat’s DMN, the anterior cingulate cortex (Nair et al., 2018). The cellular basis for the association of these two areas has been characterized with retrograde tracing studies (Chandler and Waterhouse, 2012; Chandler et al., 2013). In humans, it has been demonstrated that basal forebrain has a major functional coupling with the anterior cingulate cortex and with ventromedial prefrontal cortex (Markello et al., 2018). We speculate that the putative differences in the basal forebrain projections may be one explanation for the more diffuse DMN activation at the midline level in rats, when compared to humans (Lu et al., 2012).*”

G) p.14: It should be mentioned that the BF also contains GABAergic and glutamatergic populations of neurons with diverse projection patterns in addition to the cholinergic cells.

R: The following sentence was added in the discussion: “*The basal forebrain also contains GABAergic and glutamatergic neurons that mediate hippocampal theta synchronization through an indirect septo-hippocampal pathway (Dannenberg et al., 2015).*”

H) p.15: The considerations of the role of DMN in various disorders is a little too superficial to be useful. Can the authors focus on the putative specific role of the DMN, for example hypo- or hyperactivation in specific disorders mentioned?

R: We thank the reviewer for the suggestion. We now detail the altered connectivity and activations mentioned with respect to diseases in the text and in Table 5.

To complement the presented information, we also added a new references reviewing this topic: Whitfield-Gabrieli, S., Ford, J.M., 2012. Default mode network activity and connectivity in psychopathology. *Annu Rev Clin Psychol* 8, 49–76. <https://doi.org/10.1146/annurev-clinpsy-032511-143049>.

I) p.15: I would also encourage the authors to speculate why BF functional activations (gamma band LFP) are strongly coupled to the medial frontal cortex (i.e. cingulate) in the rat? It seems like the Cingulate is not a major target of the projection in humans? What is the most likely avenue for this BF-ACC coupling given the present data? Some thoughts on this would be appreciated.

R: We appreciate the suggestion. The sentence “*Our results are also concordant with recent neurophysiological evidence in rats about the influence of basal forebrain in the regulation of the DMN (Nair et al., 2018).*” was moved to the section of the discussion about animal models and comparative neuroscience. The following text was added: “*Namely, it has been shown that gamma-band local field potentials in the basal forebrain exerts influence on one of the hub regions of DMN in the rat, the anterior cingulate cortex (Nair et al., 2018). The cellular basis for the association of these two areas has been characterized with retrograde tracing studies (Chandler and Waterhouse, 2012; Chandler et al., 2013). In humans, it has been demonstrated that basal forebrain has a major functional coupling with the anterior cingulate cortex and with ventromedial prefrontal cortex (Markello et al., 2018). We speculate that the putative differences in the basal forebrain projections may be one explanation for the more diffuse DMN activation at the midline level in rats, when compared to humans, found by Lu and collaborators (Lu et al., 2012).*”

Reviewer 3:

A) [178] "Resting state functional MRI images were corrected for artifacts" - Which artifacts, what steps were followed? You need to provide detail here, it's not sufficient to say you passed data through a pipeline, without specifying which steps were taken. It is especially important to explain how you dealt with motion correction. Was a global signal regression done? A censoring approach? Removal of temporal derivatives? Because you use functional data to co-align subjects and then find increased functional correlations in the same sample, how do we know this improvement is not due to motion artifacts, which have been demonstrated to produce systematic spurious correlations and changes to apparent topological organization (e.g., Power et al., Neuroimg, 2012).

R: Correction, registration and seed-based connectivity analyses were performed using BCBtoolkit (Foulon et al. 2018). The following text was added to detail the method: “*Preprocessing steps in detail:*

a) fMRI images were first motion corrected using MCFLIRT (Jenkinson et al., 2002), then corrected for slice timing, smoothed with a full half width maximum equal to 1.5 times the

largest voxel dimension and finally filtered for low temporal frequencies using a gaussian-weighted local fit to a straight line. These steps are available in FEAT, as part of FSL package (Woolrich et al., 2009).

b) *fMRI* images were linearly registered to the *T1* images, and subsequently to the MNI152 template (2mm) using affine and diffeomorphic transformations (Avants et al., 2011; Klein et al., 2009). Confounding signals were discarded from *fMRI* by regressing out a series of potential confounding influences from the functional data. The confound matrix included the estimated motion parameters obtained from the previously performed motion correction, the first eigenvariate of the white matter and cerebrospinal fluid, as well as their first derivative. Eigenvariables were extracted using `fslmeants` combined with the `--eig` option. White matter and cerebrospinal fluid eigenvariables were extracted using masks based on the *T1* derived 3-classes segmentation thresholded to a probability value of 0.9, registered to the *rs-fMRI* images and binarized. Finally, the first derivative of the motion parameters, white matter and cerebrospinal fluid signal was calculated by linear convolution between their time course and a $[-1\ 0\ 1]$ vector.

c) Since the *rs-fMRI* signal can be heavily affected by motion, even following motion correction between temporally adjacent volumes (van Dijk et al., 2012), we estimated the signal fluctuation associated with motion and regressed it out from the *fMRI*. To this aim, we employed a recently developed and validated procedure based on data-driven Independent Component Analysis (ICA), termed ICA-Aroma (Pruim et al., 2015). This method performs an ICA decomposition of the data and estimates which components reflect motion-related noise in the *fMRI* signal on the basis of a robust set of spatial and temporal features. This is made possible due the distinctiveness of the motion-related components isolated by ICA on the *fMRI* signal (Salimi-khorshidi et al., 2014). This approach outperforms other methods such as the regression of the motion parameter estimates, while limiting in the same time the loss in degrees of freedom (Pruim et al., 2015). Compared to spike removal methods such as scrubbing (Power et al., 2012), ICA-Aroma has the advantage of preserving the temporal structure of the *fMRI* signal.”

B) [180] Was normalization linear or also nonlinear?

R: The normalization was linear (affine) and nonlinear (diffeomorphic). This information is now clarified in the text.

C) Section 2.2.1. is poorly written. It could be rewritten to provide a step-by-step description of how DMN distributions were computed. Also: how large were the ROIs used as seeds? Was an average time series used? Were individual voxels used? What was correlated with what? Were Pearson correlations used? Regression coefficients?

Partial correlations? More detail is required to reproduce this study. Why are medians used instead of means?

R: The ROIs that were used were previously published and made available by Gordon et al. (2016). We also added a table with the centers of gravity and the volume of each ROI in the Supplementary Materials (Table S1). In addition a surface map representing these ROIs was created (Supplementary Materials – Figure S1). The following sentence was included: *“The DMN nodes as defined by Gordon et al. are available in the Supplementary Materials (Figure S1 and Table S1, respectively)”*

We also completed the following section to better describe the method used to obtain DMN distributions: *“These 40 regions were used as seeds. The correlation map for each seed was obtained using the Funcon-Connectivity tool of the Brain Connectivity and Behaviour toolkit (<http://toolkit.bcblab.com>; Foulon et al., 2018). This tool calculates the Pearson's correlation between the mean rs-fMRI time-course of the seed with the rest of the brain to generate the functional connectivity map for every seed (Foulon et al., 2018). The DMN map of each participant was obtained by calculating the median of the 40 seed-based correlation maps of each individual using FSL (Jenkinson et al., 2012).”*

The median was chosen as the centrality measure, instead of the mean, because it is less sensitive to outliers. This is especially relevant in small samples sizes (40 maps in this case). The following sentence was included: *“The median was used, instead of the mean, because it is less sensitive to outliers, being a better centrality measure for small sample sizes (Kenney, 1939).”*

D) [197] "To obtain a group DMN map in the structural space, the median of the twenty individual maps was derived." -- Not sure what this means. What is "the structural space"? It becomes clearer later in the text, but you should introduce these terms early and not assume the reader knows what they mean. Also should be more clear that this is a "group median map".

R: We clarified the sentence: *“We then calculated the median across the DMN maps of the 20 participants to obtain the median DMN map in the MNI152 space”*.

E) [213] "different regions of interest identified in the DMN maps" -- how were these ROIs derived? Was thresholding applied, and if so what was the rationale? Were time-series averaged or was each voxel correlated separately?

R: To specify how the ROIs were derived, the following text was added: *“Cortical regions of interest were defined according to the previous anatomical models of the DMN (Andrews-Hanna et al., 2010; Buckner et al., 2008). Subcortical regions of interest were defined manually based on the experienced judgment of the neuroanatomists among the authors*

combined with the careful comparison with previously published atlases (Catani and Thiebaut de Schotten, 2012; Nieuwenhuys et al., 2008).” In addition, a map with these ROIs was now made available in the Supplementary Materials (Figure S2).

We applied a threshold of 0.3 because it corresponds to a medium effect size. The rationale for this choice was now added to this section: “*For the functional and structural-based DMN maps, a prespecified threshold of 0.3 was applied. This value was chosen because it corresponds to a medium effect size (Cohen, 1992).*”

For the analysis, the time-series were averaged. We now clarified this point in the text: “*Time-series of rs-fMRI of each individual in the different regions of interest identified in the DMN maps were extracted and the averaged value of the signal across voxels of each region was obtained, using the command fslstats (with the options -k and -M)*”.

F) [220] "p-value corrected for multiple comparisons at a threshold of <0.0001 (Bonferroni correction)." -- Does this mean there were 500 comparisons?

R: This corrected p-value corresponds to 528 comparisons. Since we have 33 regions of interest, the number of correlations was $33 \times 32 / 2$.

G) [231] "A percentage of volume overlap between the DMN map and each nucleus of interest" -- Were the volumes comparable? What is this quantity meant to show? How dependent is this quantity on the threshold used to define the DMN map (which hasn't been specified)? The rationale for thresholding becomes even more important since you are using these values to draw inferences...

R: The threshold used was the same and for the same reasons specified in E). We have chosen to present percentage to account for the differences in the volumes of the nuclei. For the purpose of transparency, an additional column was inserted in the Table 3 with the absolute volumes of intersection. The legend of this table was revised accordingly: “*Proportion of individual DMN maps intersecting, average proportion of the intersection and average volume of the intersection for each nucleus.*”

In addition, the density map of proportion of individuals overlapping the hypothesized regions presented in Figure 4 was made available at neurovault.org (<https://neurovault.org/collections/CTTXXAYJ/>).

H) [237] Why use a deterministic tractography approach instead of probabilistic? How do you account for variability in streamline distribution? Does this method account for crossing fibres or different anisotropy profiles generally?

R: We used a deterministic approach due to its proven reliability with anatomical studies (Catani et al., 2012; Dell'Acqua et al., 2013; Karolis et al., 2019; Thiebaut de Schotten et al.,

2011). We agree that tractography has potential limitations. To limit their impact we used protocols that have shown reliable results with axonal tracing studies and with postmortem dissections. To acknowledge the limitations of the tractography analysis, two references were added to the manuscript: “Jones, D.K., Cercignani, M., 2010. Twenty-five pitfalls in the analysis of diffusion MRI Data. *NMR Biomed* 23, 803–820” and “Maier-Hein, K.H., Neher, P.F., Descoteaux, M., 2017. The challenge of mapping the human connectome based on diffusion tractography. *Nat Commun* 8, 1349–136”. In addition, the following paragraph was added: “*Finally, tractography analyses can produce inaccurate results (Jones and Cercignani, 2010; Maier-Hein et al., 2017). To avoid these caveats, we employed methods that have previously demonstrated high anatomical reliability when compared to axonal tracing and with post-mortem dissections (Catani et al., 2012; Dell’Acqua et al., 2013; Karolis et al., 2019; Thiebaut de Schotten et al., 2011)*”.

To account for the variability of streamline distribution across individuals, we converted individual tracts into volumes (Thiebaut de Schotten, et al. 2011). Then, group representative tracts were obtained by applying one-sample T-test. We now report these statistical maps in the Supplementary Materials – Figure S7. In addition, the following paragraph was modified to clarify this point: “*In order to have a group-representative tract volume for each tract, individual tracts were converted into volumes using tckmap command of MRtrix toolbox (<http://www.mrtrix.org/>; Calamante et al., 2010). Each voxel was binarized in 1 or 0 according to being or not being intersected by a streamline (Thiebaut de Schotten et al., 2011b). Lastly, a one sample T-test was calculated using FSL randomise, with variance smoothing of 4mm (Thiebaut de Schotten et al., 2011b). BrainVisa was used to create the corresponding illustration of the tracts that reached significance (Rivière et al., 2011).*” The sentence “*Statistical maps of the tractography analysis are presented in the Supplementary Materials (Figure S7).*” was also included. The reference “Thiebaut de Schotten, M., et al. 2011. Atlasing location, asymmetry and inter-subject variability of white matter tracts in the human brain with MR diffusion tractography. *Neuroimage* 54, 49–59.” was added.

I) [250] "Regions of interest were defined according to the previous anatomical models of the DMN" -- It should be explained how these were derived, what their granularity is (and how many there are), why they were chosen over other alternatives, etc. More detail on these ROIs is required as they are a critical aspect of this method.

R: We are happy to provide more details as well as test another alternative.

The definition of the cortical regions of interest was made according to the literature (Buckner et al., 2008; Andrews-Hanna et al., 2010). However, previous models do not include subcortical structures. Therefore, these regions were drawn manually based on the experienced judgment of the neuroanatomists among the authors combined with the careful

comparison with previously published atlases (Nieuwenhuys et al. 2008; Catani and Thiebaut de Schotten 2012). To clarify this aspect, the following paragraph was reformulated: *“Cortical regions of interest were defined according to previous anatomical models of the DMN (Andrews-Hanna et al., 2010; Buckner et al., 2008). Subcortical regions of interest were defined manually based on the experienced judgment of the neuroanatomists among the authors combined with the careful comparison with previously published atlases (Nieuwenhuys et al. 2008; Catani and Thiebaut de Schotten 2012).”* In addition, a map with these ROIs was now made available in the Supplementary Materials (Figure S2).

The threshold used for the anatomical delimitation was a Pearson correlation of 0.3, because it corresponds to a medium effect size. This point was now clarified accordingly in the text: *“For the functional and structural-based DMN maps, a prespecified threshold of 0.3 was applied. This value was chosen because it corresponds to a medium effect size (Cohen, 1992).”*,

The section about the resulting regions of interest was also appended: *“Twenty-four regions of interest were defined based on the DMN we obtained in the functional space and concordant with the previous anatomical models of the DMN (Andrews-Hanna et al., 2010; Buckner et al., 2008). Nine additional regions were defined as being solely revealed after the functional alignment. These regions included the left and right thalamus, the left and right basal forebrain, the midbrain, the left and right ventral lateral prefrontal cortex (VLPFC) and the left and right caudate nucleus (inferior regions of the nuclei), resulting in a total of 33 regions of interest.”*

An alternative method to define the cortical regions of interest is to use the DMN parcellation proposed by Gordon et al. We repeated the analysis using these areas. The results are presented in the Supplementary Materials – Figure S4. Accordingly, the following sentences were added to the methods and to the results section respectively: *“Two alternative correlation matrices using cortical DMN areas according to Gordon et al. (2016), instead of the ones defined in classical anatomical models of Andrews-Hanna et al. (2010) and Buckner et al. (2008), were calculated.”*. *“The alternative correlation matrices using cortical DMN areas according to Gordon et al. (2016) are presented in the Supplementary Materials (Figure S4).”*

J) [253] "The command tckedit of MRtrix toolbox (<http://www.mrtrix.org/>; Tournier et al., 2012) was used to isolate the tracts of interest." -- What does it mean to "isolate the tracts of interest"? How does tckedit do this?

R: We used the command tckedit to select group of streamlines from the tractography. The “-include” arguments were composed by combinations of DMN regions of interest. With this,

we only extracted streamlines that belonged to a certain white matter tract and that crossed at least two DMN regions of interest.

This information was added to the manuscript: *“The –include arguments were composed by combinations of DMN regions of interest to guarantee that the selected streamlines crossed at least two DMN regions of interest.”*

K) [256] "In order to have a group-representative tract volume for each tract, one sample T-test of individual tract volumes was calculated using FSL randomise" -- How are "tract volumes" computed? What does this computation depend on? What does it represent in biological terms?

R: Tracts were converted into maps using the `tckmap` command of MRtrix toolbox (<http://www.mrtrix.org/>; Calamante et al., 2010). Each voxel was then assigned a value of 1 if it intersected a streamline and 0 otherwise (Thiebaut de Schotten et al., 2011). Then, we applied a one sample T-test to the group of individual tract maps using FSL `randomize`, with a variance smoothing of 4mm, in order to determine which voxels reached statistical significance. The method is now clarified as follows: *“In order to have a group-representative map for each tract, individual tracts were converted into maps using `tckmap` command of MRtrix toolbox (<http://www.mrtrix.org/>; Calamante et al., 2010). Each voxel was binarized in 1 or 0 according to being or not being intersected by a streamline (Thiebaut de Schotten et al., 2011b). Lastly, a one sample T-test was calculated using FSL `randomise`, with variance smoothing of 4mm (Thiebaut de Schotten et al., 2011b). BrainVisa was used to create the corresponding illustration of the tracts that reached significance (Rivière et al., 2011).”*

As previously stated in commentary H), we also added statistical maps of this calculation in the Supplementary Materials – Figure S7.

In biological terms, it has been shown that the described method to create group-representative tract maps has a good anatomical correspondence, when compared to the histological atlas of white matter tracts. The sentence *“The resulting group-representative tract maps using this method were shown to have a good anatomical correspondence with histological atlases of white matter tracts (Bürgel et al., 2006; Thiebaut de Schotten et al., 2011b).”* was added to the manuscript.

L) [271] "The matrices were binarised (no threshold) for structural connection or no structural connection between regions." -- What does "no threshold" mean? On what basis was this binarization performed? Since you are not using probabilistic tractography, and have no estimate of uncertainty how are you confident this is the "correct" connectome (i.e., no false negatives)? Particularly important since you only

have 20 subjects. Would adding more subjects change this connectome (and its topological characterization)?

R: The cells of the matrices represented the number of streamlines connecting the regions of interest. Each cell was binarized depending on the existence (value 1) or not (value 0) of any streamline connecting two regions of interest. The binarization was performed mainly because the streamline count is a very biased measure of connection strength (Gong et al., 2009) and is not a good measure of the true number of axonal projections (Jones et al., 2013). In addition, previous work has shown that using different thresholds in terms of streamline count (1, 3 or 5) for binarisation does not change the overall results of the network analysis (Shu et al., 2011). As it has been used in other works studying structural connectivity with a graph theory approach (Gong et al. 2009; Shu et al., 2011), we followed a protocol that only considers streamlines ending in the regions of interest.

To better explain this point, the following text was changed and the corresponding references included: *“The matrices were binarised depending on the existence or absence of streamlines connecting two regions of interest (Gong et al., 2009; Shu et al., 2011). The defined threshold for binarisation was 1 because the number of streamlines does not reflect the connectivity strength or the true number of axonal projections between two brain regions (Gong et al., 2009; Jones et al., 2013), and previous evidence has shown that changing the streamline count threshold for binarisation (between 1 and 5) does not change the overall results of the network analysis.”*

We are aware of the limitations of tractography analysis, and have further detailed the potential caveats of the tractography analysis in the limitation section.

M) Fig 2d: Higher FC for functionally aligned maps: You use the same subjects to functionally co-align as to compute FC. FC is used as the feature for coalignment. Doesn't it seem trivial (i.e., double-dipping) then that you find stronger FC for the functional versus structural maps? It would be more convincing had you shown this for an independent sample. This should be discussed.

R: In fact, demonstrating higher functional connectivity after performing functional alignment is circular. However, we originally performed this analysis to demonstrate that the method of functional alignment was valid at the subcortical level. We clarified this point in the limitation section of the manuscript: *“Furthermore, demonstrating that the additional step of functional alignment results in general higher functional correlations is somewhat circular. However, our purpose was to demonstrate that this increase in correlation revealed subcortical structures that were previously neglected in the literature.”*

N) [305] "statistically significant in 95% and in 18% of pairs, whether or not correcting and correcting for multiple comparisons was applied" -- If you correct for multiple comparisons (which you should), there's no need to report uncorrected statistics. [Also, the sentence is poorly constructed].

R: The uncorrected statistics were removed, both in text and in the Figure 3b. The sentence was reformulated to: *"The difference was statistically significant in 18% of pairs after Bonferroni correction for multiple comparisons (p -value <0.0001)."*

O) [306] "Table 2 represent the MNI coordinates of all the regions of interest in the DMN map" -- Should specify that these coordinates are the center of mass of the ROI.

R: The sentence was changed to: *"Table 2 represents the MNI coordinates of the centres of gravity of all the regions of interest in the DMN map"*.

P) Fig 3b is confusing because it is not clear whether you are showing partial correlation coefficients or differences in these coefficients between structural and functional maps. The significance levels appear to refer to the latter, but the caption indicates these are coefficients only for the functional map. Why use asterisks, instead of changing the edge colours to indicate significance? Why show uncorrected significance, when you clearly need to apply a FWE correction here? Also, the labels are not all readable.

R) The figure represents partial correlations. To clarify this point the legend was revised as follows: *"b) graph representation of the partial correlations between regions of interest in the functional space (connections with partial correlation above 0.2 are depicted; darker grey tones represent stronger connections)"*. We revised the figure so that only statistically significant partial correlations after correction for multiple comparisons are shown. They are represented with dashed edges. The legend was changed accordingly: *"Statistically significant partial correlations have a dashed borderline (one-sample T-test; $p<0.0001$, which corresponds to the Bonferroni corrected level of significance)."*

The font size of the labels was also increased.

Q) Fig 4: The colour map here has red/orange on both ends of the scale, which makes interpretation difficult. This should be fixed to have unique (preferably hot/cool) colours for each end of the scale. Also, units would make more sense as percentages, especially since these are discussed in the corresponding text.

R: The colour map was changed according to the reviewer's recommendations. In addition, the units were transformed into percentages.

R) The percent overlap metric relies on a threshold, and how this threshold was determined is unclear (statistical significance at 0.05? corrected/uncorrected? FWE? Cluster-level?). Instead of a (necessarily) arbitrary threshold approach, why not simply compute the mean/std correlation values for these voxels? Surely this would highlight the overlapping areas in a less binarized fashion...?

R: The threshold used to determine the overlap percentage was a Pearson's correlation = 0.3 because it corresponds to a medium effect size.. This point was now clarified in the text: *"The same threshold of a Pearson's correlation of 0.3 was applied for this analysis, as specified in 2.2.3."*

As requested by the reviewer, we also added mean and standard deviation maps of the correlation values: *"Maps of the mean and standard deviation Pearson's correlation of the new DMN map after registration to the MNI space are available in the Supplementary Materials (Figures S5 and S6, respectively)."*

S) [329] "nine additional regions of interest were defined including..." -- how were these defined? Was a threshold set? These details are important!

R: These nine additional regions were those found in the functionally aligned group map but not in the structurally aligned group map. We now made this information explicit in the manuscript: *"Nine additional regions were defined as being solely revealed after the functional alignment. These regions included the left and right thalamus, the left and right basal forebrain, the midbrain, the left and right ventral lateral prefrontal cortex (VLPFC) and the left and right caudate nucleus (inferior regions of the nuclei), resulting in a total of 33 regions of interest."* A map with these ROIs was now made available in the Supplementary Materials (Figure S2).

We also detailed how the subcortical regions were defined: *"Subcortical regions of interest were defined manually based on the experienced judgment of the neuroanatomists among the authors combined with the careful comparison with previously published atlases (Catani and Thiebaut de Schotten, 2012; Nieuwenhuys et al., 2008)."*

The threshold was a Pearson's correlation of 0.3: *"For the functional and structural-based DMN maps, a prespecified threshold of 0.3 was applied. This value was chosen because it corresponds to a medium effect size (Cohen, 1992)."*

T) [447] "Hence, integrating present and previous findings, it appears that the DMN, as defined by functional connectivity, is at the interplay between a cholinergic and a dopaminergic system dedicated to memory and emotion." -- It is not clear how the present findings elucidate this system, and neurotransmitter systems in particular, beyond what we already knew. The methods used here do not allow the level of

resolution (temporal or spatial) to permit any sort of direct inference with respect to these systems. It is not clear what "at the interplay" refers to. I'd suggest removing these speculative statements, or make it clearer that they are purely speculative and not in any way supported by the evidence presented in this study. It is important to put your work in context, but equally important to avoid unwarranted conclusions.

R: We completely agree with the reviewer. The point made is purely speculative and it was mentioned only to put our research in context and to foster future research.

Therefore, the paragraph was reformulated to reinforce its speculative nature: *“Hence, combining present and previous findings, we speculate that the DMN, as defined by functional connectivity, might have a putative role in the integration of cholinergic and dopaminergic systems dedicated to memory and emotion.”*

REVIEWERS' COMMENTS:

Reviewer #1 (Remarks to the Author):

The authors have adequately addressed my comments and I am now happy to recommend publication.

Reviewer #2 (Remarks to the Author):

I have carefully reviewed the revisions performed by the authors. My comments had not identified any major problems but mostly concerned various clarifications and requested additional details. The authors have provided all information in a satisfactory manner and i recommend acceptance of this manuscript.

Reviewer #3 (Remarks to the Author):

The authors have put a considerable amount of effort into addressing the reviewer comments. While the methodology isn't perfect, it is much better explained, and limitations are now acknowledged and discussed. I am satisfied with their response and don't have anything major to add.